# Precise recognition of benzonitrile derivatives with supramolecular macrocycle of phosphorylated cavitand by co-crystallization method

Heng Li [1], Zhijin Li[1], Chen Lin [1] ✉, Juli Jiang [1] ✉ & Leyong Wang [1]

The importance of molecular docking in drug discovery lies in the precise recognition between potential drug compounds and their target receptors, which is generally based on the computational method. However, it will become quite interesting if the rigid cavity structure of supramolecular macrocycles can precisely recognize a series of guests with specific fragments by mimicking molecular docking through co-crystallization experiments. Herein, we report a phenylphosphine oxide-bridged aromatic supramolecular macrocycle, F[3]A1-[P(O)Ph]$_3$, which precisely recognizes benzonitrile derivatives through non-covalent interactions to form key-lock complexes by co-crystallization method. A total of 15 various benzonitrile derivatives as guest molecules are specifically bound by F[3]A1-[P(O)Ph]$_3$ in co-crystal structures, respectively. Notably, among them, crisaborole (anti-dermatitis) and alectinib (anti-cancer) with the benzonitrile fragment, which are two commercial drug molecules approved by the U.S. Food and Drug Administration (FDA), could also form a key-lock complex with F[3]A1-[P(O)Ph]$_3$ in the crystal state, respectively.

Biological receptors[1,2] possess specific binding sites for high-affinity substrates, which are crucial for maintaining normal biological activity. Molecular docking technology[3,4] serves as a pivotal methodology in drug design, facilitating the characterization of the receptor and elucidating the intricate interactions between the receptor and the drug molecule, and it plays a critical role in the precise binding of biological receptors to substrates containing structurally identical fragments, establishing a specific lock-and-key recognition relationship[5,6]. It enables the study and prediction of the binding process, facilitating the development and optimization of drugs for disease treatment[7,8], and biological regulation[9,10]. The nature of molecular docking between a biological receptor and a specific substrate involves a molecular recognition process[11–16] that is primarily driven by the specific structural and chemical properties of biological receptors: (a) the biological

receptor typically has a cavity shape, providing an ideal environment for substrate binding and (b) amino acid residues[17–19] within the biological receptor cavity efficiently interact with functional groups of the substrate through non-covalent interactions such as hydrogen bonding[20–22]. Inspired by biological receptors, numerous supramolecular macrocycles were studied through conventional molecular docking by using computational and simulation methods[23–27]. Prominent examples of such supramolecular macrocycles include cucurbiturils[28–33] and pillar[n]arenes[34–39], which exhibit exceptional host-guest binding capabilities due to their well-defined and complementary structures that can effectively encapsulate and interact with specific guest molecules. The identification of a specific binding mode by molecular docking based on the computational method elucidated structural compatibility between supramolecular

[1]State Key Laboratory of Analytical Chemistry for Life Science, Jiangsu Key Laboratory of Advanced Organic Materials, School of Chemistry and Chemical Engineering, Nanjing University, 210023 Nanjing, China. ✉e-mail: linchen@nju.edu.cn; jjl@nju.edu.cn

macrocycles and guests. However, the study of the precise recognition of a series of guests with specific fragments in the solid phase, as molecular docking does with supramolecular macrocycles, is rare in supramolecular chemistry. Therefore, if the rigid cavity structure of supramolecular macrocycles can precisely recognize a series of guests with specific fragments by mimicking molecular docking through co-crystallization experiments, not only spatial and energetic complementarity with the target guest molecule can be achieved, but also the precision and reliability of this specific binding capability can be greatly improved.

Herein, in this work, a rigid supramolecular macrocycle was designed and synthesized, named F[3]A1-[P(O)Ph]₃, in which the bridged phenylphosphine oxide groups were introduced at the middle rim of 2,7-OH-F[3]A1[40], resulting in the formation of a completely locked conformation of the macrocycle. F[3]A1-[P(O)Ph]₃ has a triangular rigid cavity that makes itself easily crystalize and allows the possible formation of a key-lock complex with the guest molecules. In addition, the lower rim of F[3]A1-[P(O)Ph]₃ contains multiple hydrogen bond donors, leading to the formation of a distinct partially positively charged triangular region at the lower rim of the macrocycle. Therefore, based on the structural features and electron density distribution of the above supramolecular macrocycle, guest molecules with benzene rings and partially negatively charged molecular terminals in their structure were selected for specific binding study through general recognition based on the co-crystallization method with F[3]A1-[P(O)Ph]₃ (Fig. 1a, b). It was found that precisely recognized with guest molecules, the key-lock co-crystal structures of F[3]A1-[P(O)Ph]₃ with benzonitrile derivatives were obtained, typically, including the commercial drugs as guest molecules, crisaborole (anti-dermatitis) and alectinib (anti-cancer), approved by the U.S. Food and Drug Administration (FDA) in 2016 and 2015, respectively (Fig. 1c). It turned out that the precise recognition based on the co-crystallization method could

be highly efficient, resulting in the formation of key-lock complexes in the solid state. The research can open up exciting possibilities for determining the structures of drugs containing benzonitrile fragments.

## Results and discussion

F[3]A1-[P(O)Ph]₃, a phenylphosphine oxide-bridged aromatic macrocycle, was synthesized simply by stirring the macrocycle compounds 2,7-OH-F[3]A1[40] and PhCl₂P in pyridine at 80 °C for 24 h followed by the addition of H₂O₂ to oxidize the intermediates (Fig. 2a). Full details of the synthesis are provided in the Supplementary Fig. 1. ¹H NMR (Supplementary Fig. 10) and ¹³C NMR (Supplementary Fig. 11) demonstrated the successful preparation of the F[3]A1-[P(O)Ph]₃. To further validate the structure, we rapidly obtained high-quality single crystals of F[3]A1-[P(O)Ph]₃ within a day by evaporating a saturated dichloromethane solution, which demonstrated the ease of F[3]A1-[P(O)Ph]₃ with the rigid cavity structure to readily crystallize. The single crystal data show that the diameters of the upper and lower rims of F[3]A1-[P(O)Ph]₃ are approximately 10.3 and 5.6 Å, respectively, and the depth of the cavity is 6.8 Å (Supplementary Fig. 15). F[3]A1-[P(O)Ph]₃ with a triangular structure has rim lengths of 11.2 × 11.2 × 11.6 Å and apex angles of 108°, 102°, and 102°. Notably, F[3]A1-[P(O)Ph]₃ has a completely locked conformation and symmetrical formation of a rigid triangular structure, which is attributed to the introduction of phenylphosphine oxide at the middle rim of 2,7-OH-F[3]A1. The oxygen atoms in all three phenylphosphine oxide groups point upward inside the cavity, and the benzene ring extends to the outer dimensions of the cavity, presumably due to steric hindrance of the phenylphosphine oxide structure. Furthermore, the structure of F[3]A1-[P(O)Ph]₃ was further validated by the acquisition of crystal structures in toluene and benzyl alcohol solvents, providing additional evidence and confirmation of the authentic and stable conformation of the cavitand molecule (Supplementary Figs. 16, 17). These crystal structures reveal that aryl solvent molecules such as toluene and benzyl alcohol can occupy the cavity of F[3]A1-[P(O)Ph]₃ through π-π stacking interactions with the fluorenyl moiety of macrocycle ring, suggesting that the structure of F[3]A1-[P(O)Ph]₃ has the ability to encapsulate guest molecules. An important aspect to highlight is the presence of multiple hydrogen bond donors in F[3]A1-[P(O)Ph]₃. Positioned at the lower rim of the macrocycle, these ring-shaped and partially positively charged hydrogen donors have the ability to form bonds with negatively charged guest groups, establishing themselves as a truly effective lock-like structure (Fig. 2a). Such interactions facilitated by hydrogen bonding prove highly advantageous in the acquisition of single crystals, augmenting the adaptability in recognizing and binding guest molecules. Additionally, we performed density functional theory

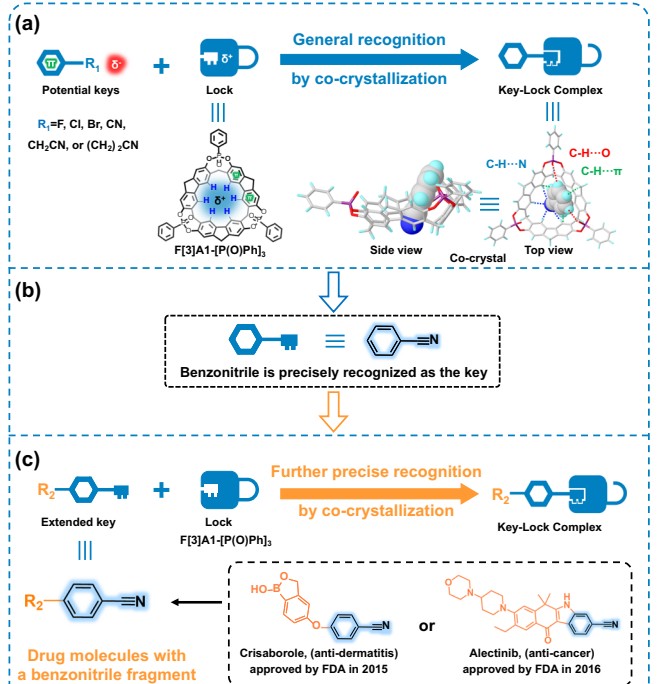

**Fig. 1 | Precise recognition studies of F[3]A1-[P(O)Ph]₃ with guest molecules by co-crystallization to form key-lock complexes. a** The potential guest molecules were generally recognized. **b** Benzonitrile was precisely recognized as the key. **c** More complicated guest molecules with a benzonitrile fragment, including drug molecules crisaborole (anti-dermatitis) and alectinib (anti-cancer), were further precise recognition.

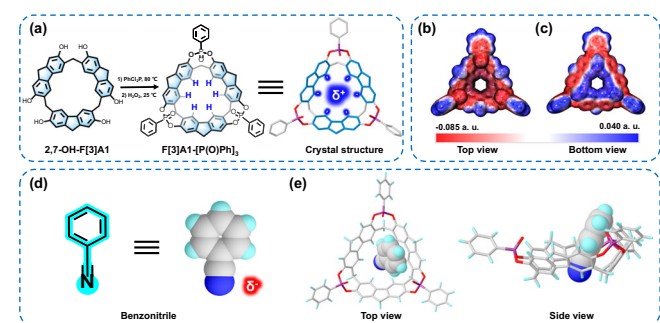

**Fig. 2 | Synthesis of F[3]A1-[P(O)Ph]₃ and general recognition with benzonitrile by co-crystallization. a** Synthetic scheme and single crystal structure of F[3]A1-[P(O)Ph]₃. **b** Top view and **c** bottom view of the calculated electrostatic potential (ESP) maps of F[3]A1-[P(O)Ph]₃. **d** Chemical structure of benzonitrile. **e** Top and side views of the crystal structure of benzonitrile forming a key-lock complex with F[3]A1-[P(O)Ph]₃.

calculations (DFT) at the B3LYP/6-31G(d) level of theory[41], incorporating the em = gd3 dispersion correction[42], specifically to illustrate the electrostatic potential (ESP) map of F[3]A1-[P(O)Ph]₃ in its ground state, which reveals that the cavity of F[3]A1-[P(O)Ph]₃ exhibits a negatively charged region (Fig. 2b and Supplementary Fig. 13). Importantly, the multiple hydrogen atoms located towards the lower rims of F[3]A1-[P(O)Ph]₃ show a distinct triangular region of positive potential. In the ESP calculations, the surface maximum of this positive potential reaches up to 0.034 eV (Fig. 2c). The calculations are consistent with previous predictions for the presence of multiple partially positively charged hydrogen bond donors at the lower rim of the F[3]A1-[P(O)Ph]₃ in a lock-like structure.

With the lock-like structure of host macrocycle F[3]A1-[P(O)Ph]₃, the precise recognition of guest molecules using the co-crystallization method aims to accelerate the identification of guest molecules with better binding properties to F[3]A1-[P(O)Ph]₃ and to rapidly determine the key-lock binding mode, with the advantage of more direct validation. Therefore, the guest molecules containing a benzene fragment ring and a partially negatively charged terminal acting as keys such as chlorobenzene, bromobenzene, *p*-fluorotoluene, benzonitrile, *p*-hydroxybenzyl cyanide, and 3-phenylpropionitrile were precisely recognized by co-crystallization method, respectively, which appears intriguing as it promises to achieve precise recognition accuracy and reliability between the host macrocycle and the guest molecule.

Consequently, their co-crystals with F[3]A1-[P(O)Ph]₃ were all successfully obtained by slow evaporation of a saturated dichloromethane solution. The solid-state co-crystal structures were then analyzed by single crystal X-ray crystallography. Interestingly, their high-resolution crystal structures show that only the benzonitrile molecule forms the key-lock complex with F[3]A1-[P(O)Ph]₃ (Figs. 1a and 3a).

To understand the key factors behind the formation of key-lock complexes between benzonitrile and F[3]A1-[P(O)Ph]₃, their high-resolution crystal structures have been carefully examined. It is shown that the benzene ring of benzonitrile engages in a π-π stacking interaction with the rigid fluorene moiety on one side of the F[3]A1-[P(O)Ph]₃, with a distance of approximately 3.4 Å between them, contributing to the stability of the key-lock complex. In addition, the terminal cyanide group (-CN) of benzonitrile, which carries a partially negative charge, forms six CH···N interactions with the ring-shaped hydrogen atoms carrying the partially positive charge located at the lower rim of the benzene ring of F[3]A1-[P(O)Ph]₃. In these interactions, the phenyl CH acts as a hydrogen bond donor, and the cyanide group acts as a hydrogen bond acceptor. The formation of these CH···N interactions further strengthens the bonding between the host macrocycle F[3]A1-[P(O)Ph]₃ and guest molecules, besides π-π stacking interactions, which firmly anchors the benzonitrile guest molecule within the macrocycle cavity. The independent gradient model based on Hirshfeld partition (IGMH)[43,44] was established to characterize the

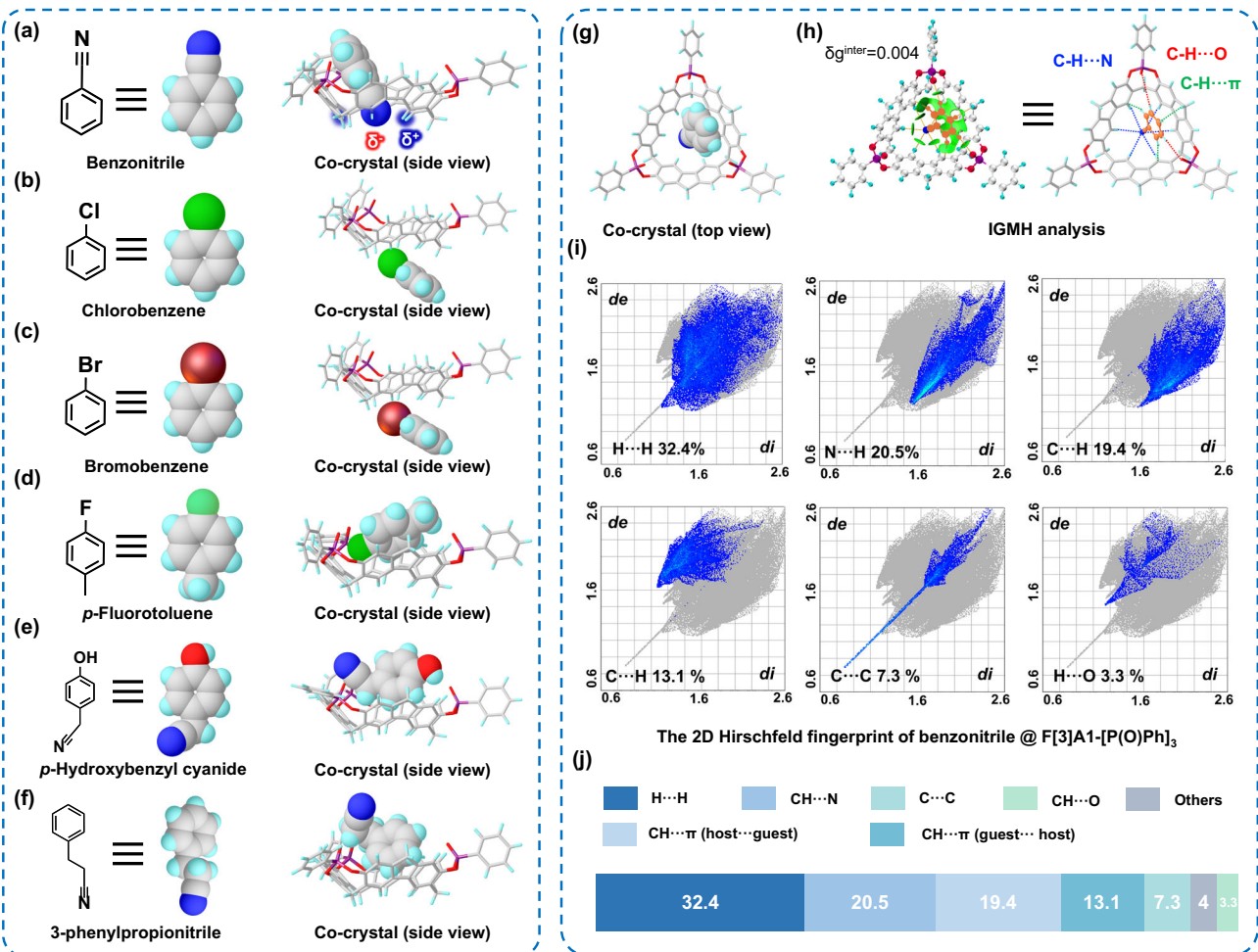

**Fig. 3 | An investigation of the formation of a key-lock complex with benzonitrile and F[3]A1-[P(O)Ph]₃.** Chemical structure of **a** benzonitrile, **b** chlorobenzene, **c** bromobenzene, **d** *p*-fluorotoluene, **e** *p*-hydroxybenzyl cyanide, and **f** 3-phenylpropionitrile. Side view of the single crystal structure of F[3]A1-[P(O)Ph]₃ with **a** benzonitrile, **b** chlorobenzene, **c** bromobenzene, **d** *p*-fluorotoluene, **e** *p*-hydroxybezyl cyanide, and **f** 3-phenylpropionitrile. **g** Top view of the single crystal structure of F[3]A1-[P(O)Ph]₃ with benzonitrile. **h** The independent gradient model based on Hirshfeld (IGMH) partition analysis of benzonitrile @ F[3]A1-[P(O)Ph]₃. **i** The 2D Hirschfeld fingerprint plots and **j** summary of the percentage contribution of different interactions of benzonitrile @ F[3]A1-[P(O)Ph]₃.

non-covalent interactions between F[3]A1-[P(O)Ph]₃ and benzonitrile. The bond critical point (BCP) and the bond diameter between them were carefully examined to gain insight into the strength of these interactions, which provides evidence of the strong interactions between F[3]A1-[P(O)Ph]₃ and benzonitrile, supporting the formation of the stable key-lock complex (Fig. 3h). In particular, the C-H···N (2.761, 2.740, 2.818, 3.049, 3.088, and 2.819 Å) at the lower rim of the macrocycle, the C-H···π (3.067, 2.962 and 3.036 Å) at the side of the macrocycle, and the C-H···O (3.578 and 2.674 Å) interactions together collectively lock the benzonitrile guest into the cavity. The 2D Hirschfeld fingerprinting of F[3]A1-[P(O)Ph]₃ and benzonitrile (Fig. 3i) further supports the importance of donor-acceptor interactions (C-H···π, C-H···O, and C-H···N) as the key driving forces for the formation of the key-lock complex (Fig. 3j)[45,46].

Except for the benzonitrile guest molecule, the other five guest molecules, chlorobenzene, bromobenzene, *p*-fluorotoluene, *p*-hydroxy-benzyl cyanide, and 3-phenylpropionitrile did not form key-lock binding modes with F[3]A1-[P(O)Ph]₃. For the single crystal structures of chlorobenzene (Fig. 3b) and bromobenzene (Fig. 3c) with F[3]A1-[P(O)Ph]₃, it was observed that chlorobenzene and bromobenzene were not located in the cavity of F[3]A1-[P(O)Ph]₃, which could be attributed to the fact that dichloromethane, as a solvent with a small molecular size, competed with chlorobenzene or bromobenzene to enter the cavities more easily (Supplementary Fig. 14). As a result, the guest molecules of chlorobenzene and bromobenzene were unable to form π-π stacking interactions well with the fluorene moiety in the inner part of the macrocycle. Additionally, an examination of the single crystal structure of *p*-fluorotoluene with F[3]A1-[P(O)Ph]₃ revealed the formation of a π-π stacking interactions between *p*-fluorotoluene and one side of the host macrocycle. However, it was observed that *p*-fluorotoluene does not adopt the key-lock binding mode observed in the case of benzonitrile and F[3]A1-[P(O)Ph]₃ (Fig. 3d), and it probably was caused by the presence of only one fluorine atom, with a relatively short one-bond distance from the phenyl carbon to fluorine atom, which is unable to form strong interactions with the hydrogen-bonding donors at the lower rim of F[3]A1-[P(O)Ph]₃, limiting the ability of *p*-fluorotoluene to engage in the necessary interactions for binding with F[3]A1-[P(O)Ph]₃. The independent gradient model based on Hirshfeld partition analysis calculations of the co-crystals of F[3]A1-[P(O)Ph]₃ and *p*-fluorotoluene also confirmed that the shorter bond lengths of the fluorine atoms in *p*-fluorotoluene hindered the formation of a key-lock complex (Supplementary Fig. 38).

Based on the experimental data provided above, it can be speculated that the presence of a linear functional group with an appropriate bond length, such as -CN, is a prerequisite for the formation of a key-lock binding mode. This enables the establishment of multiple CH···N interactions with the hydrogens at the lower rims of the macrocyclic ring, providing enhanced binding affinity. In other words, the absence of a benzene ring and a linear functional group of appropriate bond length in the molecular structure of the guest may not provide a level of specificity and sufficient binding strength. To further confirm the speculations discussed above, guest molecules with different lengths of cyanide-based fragments on the benzene ring, such as *p*-hydroxybenzyl cyanide and 3-phenylpropionitrile, were used to specifically bind to F[3]A1-[P(O)Ph]₃ by using co-crystallization method. As expected, the high-resolution crystal structure of *p*-hydroxybenzyl cyanide with F[3]A1-[P(O)Ph]₃ showed that although the guest molecules formed a π-π stacking interactions with the interior of F[3]A1-[P(O)Ph]₃, the cyanide-based fragments were partially oriented toward the upper rim of F[3]A1[P(O)Ph]₃ (Fig. 3e). This phenomenon can be attributed to the inability to form the non-covalent interaction with the hydrogens at the lower rim of F[3]A1-[P(O)Ph]₃ due to the longer length of the acetonitrile fragment of guest than nitrile fragment as well as the spatial site resistance. Similarly, in

the case of the guest molecule 3-phenylpropionitrile, the propionitrile fragment was also oriented toward the upper rims of F[3]A1-[P(O)Ph]₃ (Fig. 3f). This observation further supports the inference that the length of the cyanide-based fragments affects the key-lock binding mode and orientation of the guest molecule in the cavity. The formation of the key-lock complex is driven by the synergistic contribution of both π-π stacking interactions and hydrogen bonding interactions between benzonitrile and F[3]A1-[P(O)Ph]₃.

Therefore, in order to gain a more comprehensive understanding of the binding mode between benzonitrile and F[3]A1-[P(O)Ph]₃, an in-depth investigation of the host-guest binding constants was performed. ¹H NMR spectra of the titration experiment were carried out in dichloromethane-*d* (Supplementary Figs. 42, 43). In this experiment, the concentration of F[3]A1-[P(O)Ph]₃ in the solution was kept constant at 3 mM, while the concentration of benzonitrile was gradually varied from 0.4 to 11.2 mM, and the changes in the chemical shifts of specific protons of F[3]A1-[P(O)Ph]₃ could be observed (Fig. 4a). The proton $H_a$, located on the benzene ring at the lower rim of F[3]A1-[P(O)Ph]₃, experienced a chemical shift from 7.972 to 8.276 ppm. This shift of 0.304 ppm is attributed to the hydrogen bonding interactions between proton $H_a$ and the cyanide group of benzonitrile. Additionally, the protons $H_e$ and $H_f$ on the methylene group (-CH₂) at the lower rim of F[3]A1-[P(O)Ph]₃ were also affected by the presence of the cyanide group (-CN). They split from a single peak into two double peaks, with a displacement of 0.288 ppm. On the other hand, the protons $H_b$, $H_c$, and $H_d$ located at the upper rim of F[3]A1-[P(O)Ph]₃ underwent only minor chemical shift changes, which is attributed to donor and receptor interactions between the host macrocycle and guest molecules. Based on the ¹H NMR spectra of the titration experiment of $H_a$, the binding constant $K_a$, $4.153 \times 10^3$ M⁻¹, between them was obtained (Fig. 4b). The job plot experiment between F[3]A1-[P(O)Ph]₃ and benzonitrile by ¹H NMR spectra was also conducted (Fig. 4c). The job plot analysis revealed that the stoichiometric ratio of the complex formed between F[3]A1-[P(O)Ph]₃ and benzonitrile in dichloromethane-*d* was determined to be 1:1 (Fig. 4d). Binding constants between F[3]A1-[P(O)Ph]₃ and other guests were also determined (Supplementary Figs. 44–58).

The host-guest interactions observed between F[3]A1-[P(O)Ph]₃ and benzonitrile prompted further exploration of more complex derivatives of benzonitrile for precise recognition by the co-crystallization method. Therefore, the precise recognition of F[3]A1-[P(O)Ph]₃ with a variety of benzonitrile derivatives (G2-G12) has been carried out by using the co-crystallization method (Fig. 5a–k). By slow evaporation of dichloromethane solvent, their co-crystals were all successfully obtained, respectively. Consistent with the binding mode observed in the case of benzonitrile with F[3]A1-[P(O)Ph]₃, all of the guest molecules (G2-G12) formed key-lock complexes with F[3]A1-[P(O)Ph]₃, respectively. This result demonstrates the versatility and adaptability of the host molecule F[3]A1-[P(O)Ph]₃ as a lock, which is able to accommodate structurally diverse benzonitrile derivatives to form key-lock complexes. Thus, the precise recognition based on the co-crystallization method between the host macrocycle F[3]A1-[P(O)Ph]₃ and benzonitrile derivatives shows exceptional efficiency for the research of key-lock complexes. The strong interaction between the benzonitrile fragment and F[3]A1-[P(O)Ph]₃ prompted us to explore the use of a guest molecule, 1,4-bis(4-cyanostyryl)benzene (G13), which contains benzonitrile fragments at molecular both terminals to conduct precise recognition with two host macrocycles in order to achieve a binding mode with a stoichiometric ratio of 1:2 between the guest and host. As anticipated, the binding mode where two host macrocycles are connected by one guest molecule has been successfully achieved, evidenced by the single crystal structure (Fig. 5l). Correlation-independent gradient model based on Hirshfeld partition analysis (Supplementary Fig. 39) and the 2D Hirschfeld fingerprints between G13 and F[3]A1-[P(O)Ph]₃ (Supplementary Fig. 70) further

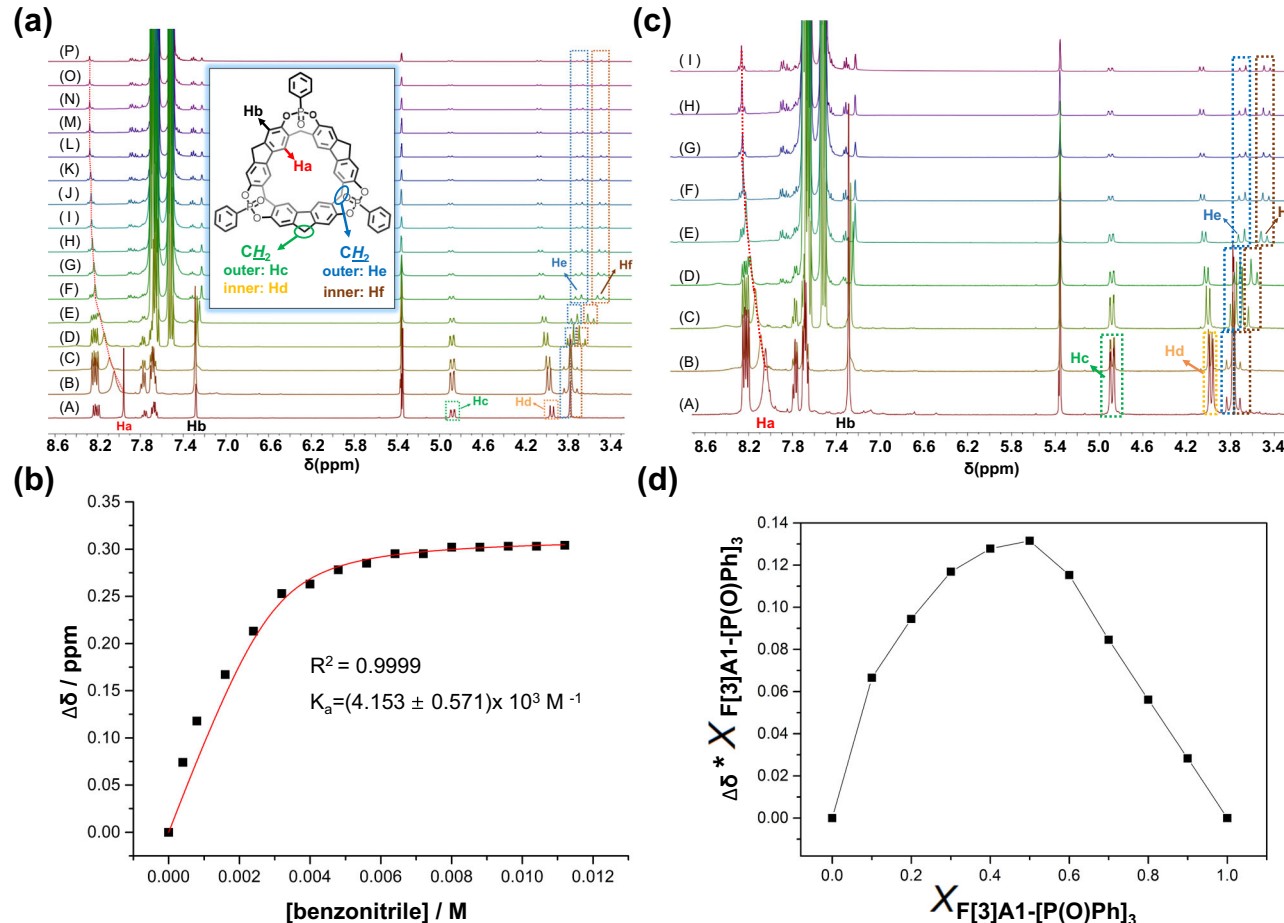

**Fig. 4 | $^1$H NMR spectra of the titration experiment of F[3]A1-[P(O)Ph]$_3$ with benzonitrile carried out in dichloromethane-*d*. a** $^1$H NMR spectra (400 MHz, CCl$_2$D$_2$, 298 K) of F[3]A1-[P(O)Ph]$_3$ at a constant concentration of 3.0 mM with different concentrations of benzonitrile (mM): (A) 0.0, (B) 0.4, (C) 0.8, (D) 1.6, (E) 2.4, (F) 3.2, (G) 4.0, (H) 4.8, (I) 5.6, (J) 6.4, (K) 7.2, (L) 8.0, (M) 8.8, (N) 9.6, (O) 10.4, and (P) 11.2. **b** The chemical shift changes of H$_a$ on F[3]A1-[P(O)Ph]$_3$ upon the addition of benzonitrile. The red solid line was obtained from the non-linear curve fitting. **c** $^1$H NMR spectra (400 MHz, CCl$_2$D$_2$, 298 K) of the concentration ratios of F[3]A1-[P(O)Ph]$_3$ to benzonitrile were shown as follows: (A) 9:1, (B) 8:2, (C)7:3, (D) 6:4, (E) 5:5, (F) 4:6, (G) 3:7, (H) 2:8, and (I) 1:9. **d** Job plot of complex F[3]A1-[P(O)Ph]$_3$ @ benzonitrile by plotting the Δδ (the chemical shift change) against the mole fraction of F[3]A1-[P(O)Ph]$_3$.

confirmed the formation of the key-lock bond pattern between the benzonitrile fragment and the host macrocycle.

The presence of benzonitrile fragments in drugs is a common occurrence, so F[3]A1-[P(O)Ph]$_3$ could possibly form enhanced key-lock binding modes with complex benzonitrile-based drug compounds, providing an experiential base for structure determination of drugs containing benzonitrile fragments. Consequently, structurally complex drug molecules containing benzonitrile fragments, such as crisaborole (anti-dermatitis, G14) and alectinib (anti-cancer, G15), were also precisely recognized for specific binding to F[3]A1-[P(O)Ph]$_3$ by the co-crystallization method. In a notable example, crisaborole[47] (G14) (Fig. 6a), a non-steroidal phosphodiesterase 4 (PDE4) inhibitor, which is a commercially available drug molecule approved by the FDA in 2016, was observed to be able to form co-crystals with F[3]A1-[P(O)Ph]$_3$ within 24 h. The high-resolution crystal structure reveals that the crisaborole molecule has successfully achieved specific binding to the host macrocycle, leading to the formation of the key-lock binding mode (Fig. 6b–d). Besides, the co-crystal of alectinib (G15) (Fig. 6f), another commercially available anticancer drug, with F[3]A1-[P(O)Ph]$_3$ was also successfully obtained as expected (Fig. 6g–i). Alectinib (G15)[48] is a potent and highly selective tyrosine kinase inhibitor that targets anaplastic lymphoma kinase (ALK), and it is commonly used in the treatment of certain types of non-small cell lung cancer (NSCLC) that have specific mutations in the ALK gene and was approved by the FDA

in 2015. The structure of the single crystal revealed a key-lock complex formation of alectinib with F[3]A1-[P(O)Ph]$_3$. Therefore, we present the high-resolution co-crystal structure of the drug molecule, alectinib. It is important to highlight that in all the crystal data obtained, the occupancy of the guest molecules is 100%. In order to delve deeper into the understanding of these interactions, DFT calculations were conducted, followed by an independent gradient model based on Hirshfeld partition analysis, as well as visualization of the BCP and bond diameter. We can clearly observe the interactions between the guest molecules G14 (Supplementary Fig. 40) or G15 (Supplementary Fig. 41) with F[3]A1-[P(O)Ph]$_3$, respectively. The 2D Hirschfeld finger-printing of G14 (Fig. 6e) or G15 (Fig. 6j) with F[3]A1-[P(O)Ph]$_3$ reveals that the key driving forces for complex formation are donor and receptor interactions (C-H···π, C-H···O, and C-H···N). All of these data strongly indicate that the binding modes of G14 or G15 with the host macrocycle are consistent with the other benzonitrile derivatives. The successful precise recognition of F[3]A1-[P(O)Ph]$_3$ with drug molecules containing benzonitrile fragments by co-crystallization method opens up possibilities for the structural determination of such drugs. Furthermore, the validation of the structural determination was meticulously carried out through a comprehensive examination of 15 guest molecules, the Oak Ridge thermal ellipsoid plot (ORTEP), and the superposition of electron density maps ($F_o$ map)[49] onto the refined outcomes (Supplementary Fig. 71), providing strong evidence for the

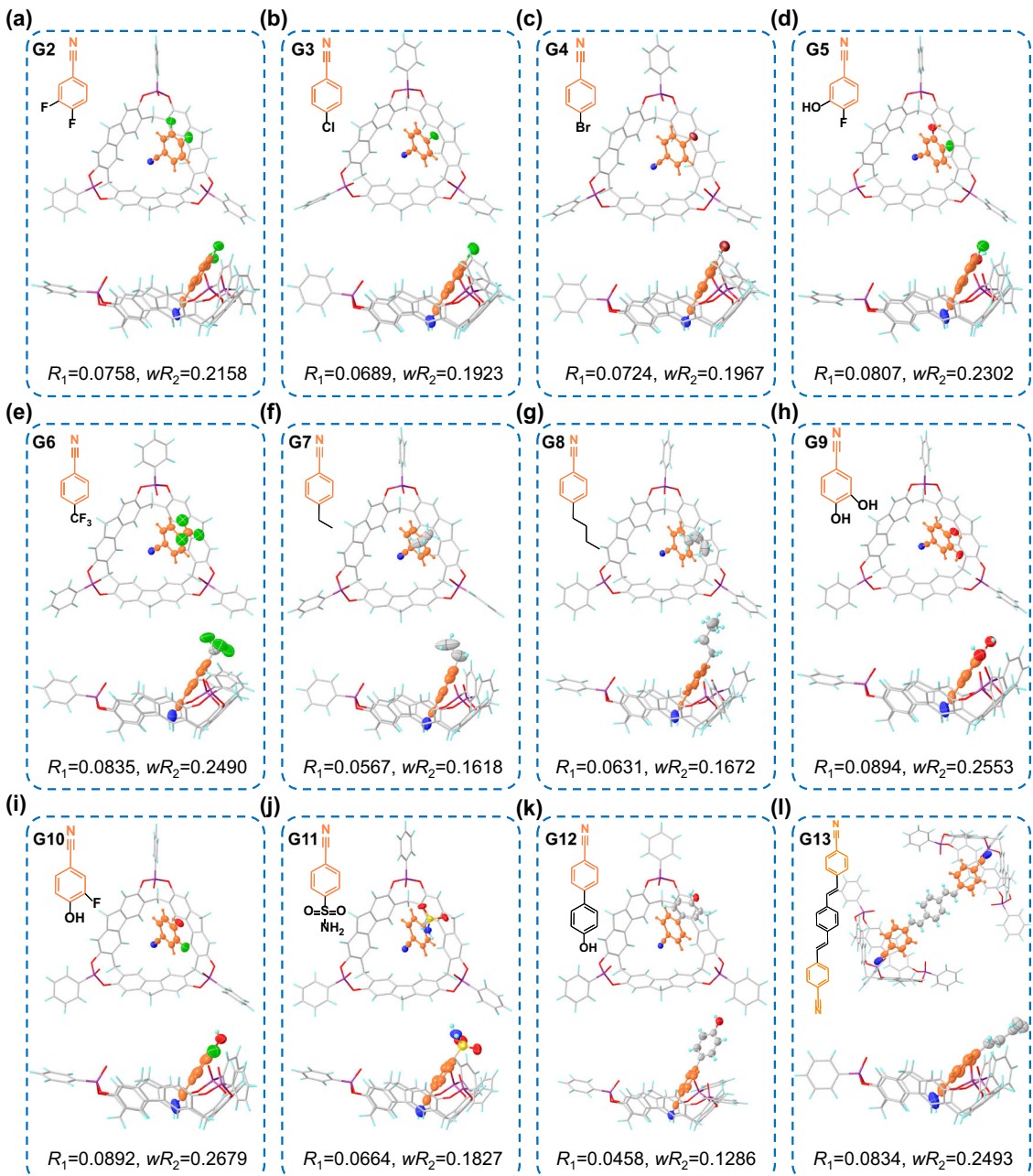

**Fig. 5 | High-resolution co-crystal structures of F[3]A1·[P(O)Ph]₃ with various benzonitrile derivatives.** Precise recognition of F[3]A1·[P(O)Ph]₃ with **a** G2, **b** G3, **c** G4, **d** G5, **e** G6, **f** G7, **g** G8, **h** G9, **i** G10; **j** G11, **k** G12, and **l** G13 by the co-crystallization method.

accuracy and reliability of the crystal data obtained from precise recognition between host macrocycles F[3]A1·[P(O)Ph]₃ and all guest molecules containing benzonitrile fragments studied above through co-crystallization methods. The success of precise recognition between supramolecular host macrocycles and guest molecules, especially with drug molecules, using the co-crystallization method, establishes a profoundly meaningful and effective experimental method, which provides an empirical basis for the structure determination of drugs containing benzonitrile fragments.

In summary, we report a phenylphosphine oxide-bridged aromatic supramolecular macrocycle F[3]A1·[P(O)Ph]₃, which has a completely locked conformation and symmetrical formation of a rigid triangular structure with multiple ring-shaped hydrogen bonding donors on the lower rim. F[3]A1·[P(O)Ph]₃ can accommodate aromatic guest molecules with a suitable size through donor and receptor interactions. In addition, the hydrogen atoms at the lower rim of F[3]

A1·[P(O)Ph]₃ form a partially positively charged region confirmed by ESP. Therefore, the guest molecules with a benzene ring in their molecular structure and a partially negatively charged terminal were successfully precisely recognized by the co-crystallization method, and it was found that the formation of a key-lock complex in the solid state between benzonitrile derivatives and F[3]A1·[P(O)Ph]₃ were obtained, which is facilitated by various non-covalent interactions. Importantly, two commercial drugs approved by the FDA, crisaborole (anti-dermatitis) and alectinib (anti-cancer), containing benzonitrile fragments, were also able to be successfully precisely recognized with F[3]A1·[P(O)Ph]₃. Impressively all benzonitrile derivatives as guest molecules demonstrated the ability to form a highly specific and complementary complex with F[3]A1·[P(O)Ph]₃, resembling a key-lock interaction, which highlights the potential and effectiveness of molecular specific recognition in determining the structure of drug molecules containing benzonitrile fragments. The studies between

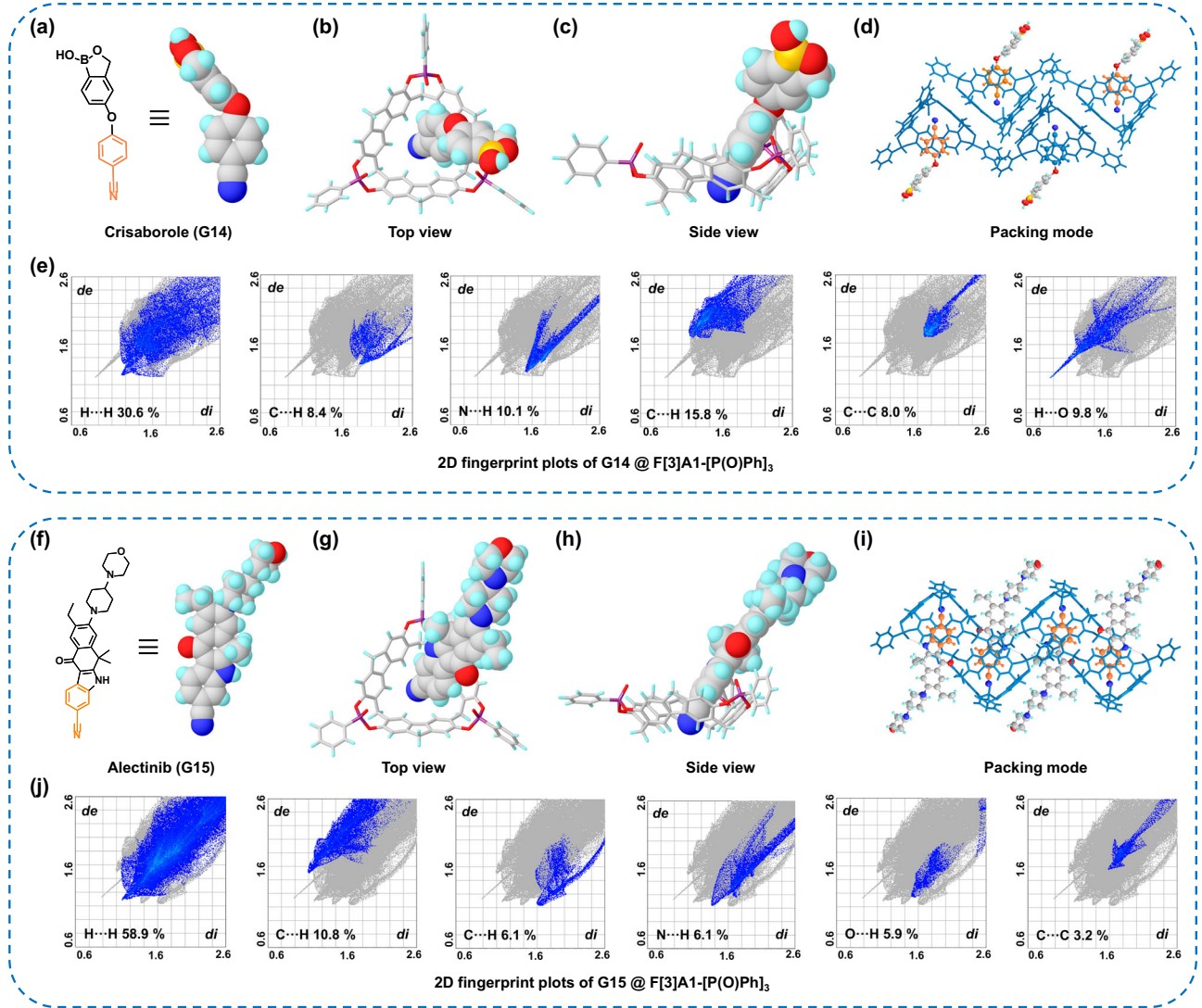

**Fig. 6 | Two FDA-approved commercially available drugs were precisely recognized with F[3]A1-[P(O)Ph]₃ by co-crystallization.** Chemical structure of the commercial drug **a** crisaborole (G14) and **f** alectinib (G15). Top view of the single crystal structure of F[3]A1-[P(O)Ph]₃ with **b** G14 and **g** G15; Side view of the single crystal structure of F[3]A1-[P(O)Ph]₃ with **c** G14 and **h** G15; Packing mode of the single crystal structure of F[3]A1-[P(O)Ph]₃ with **d** G14 and **i** G15. The 2D Hirschfeld fingerprint plots of F[3]A1-[P(O)Ph]₃ with **e** G14 and **j** G15.

supramolecular macrocycles and specific guest molecules not only contribute to our understanding of binding affinity and selectivity but also provide guidance for the design and development of supramolecular systems with enhanced selectivity and binding properties by precise recognition in the solid state based on the co-crystallization method.

## Methods

### Materials
All reactions were performed in an air atmosphere unless otherwise stated. Deuterium solvents were purchased from Aldrich. All other reagents were obtained from commercial sources and were used without further purification unless indicated otherwise. All yields were given as isolated yields. ¹H NMR and ¹³C NMR spectra were recorded on a BRUKER AVANCE III 400 MHz, and the chemical shifts (δ) for ¹H NMR spectra, given in ppm, are referenced to the residual proton signal of the deuterated solvent.

### Single crystal X-ray crystallography
All single crystal X-ray diffraction data were collected on a Bruker D8 Venture, Germany, at 193 K using Mo-Kα ($\lambda = 0.71073$ Å) or Cu-Kα

($\lambda = 1.54178$ Å) radiation. The crystal structure was solved and refined for all $F_2$ values using the SHELX (version 6.1) and Olex 2 (version 1.3) software packages[50]. After anisotropic refinement of all non-H atoms in the framework, the positions of the H atoms were calculated geometrically with riding models. The detailed experimental parameters are summarized in Supplementary Figs. 14–37.

### Theoretical and computational method
The geometries of all the ground state co-crystals were selected from the corresponding X-ray single-crystal diffraction data. All calculations were performed using the Gaussian 09 software package[41,42]. All the atomic coordinates datasets of optimized computational models are shown in Supplementary Information. The calculated electrostatic potential maps were calculated with the popular functional B3LYP/6-31G(d). The independent gradient model based on Hirshfeld partition analysis was visualized by VMD program assisted by the Multiwfn program.

### Data availability
The X-ray crystallographic coordinates for structures reported in this study have been deposited at the Cambridge Crystallographic Data

Centre (CCDC), under deposition numbers 2313215-2313217; 2313219-2313221; 2313225; 2313229; 2313233-2313234; 2313236; 2349280-2349285; 2349287-2349288; 2349290-2349293. These data can be obtained free of charge from The Cambridge Crystallographic Data Centre via www.ccdc.cam.ac.uk/data_request/cif. The authors declare that the data supporting the findings of this study are available within the paper and its Supplementary Information. The additional data can be obtained from the corresponding author. Source data are provided with this paper.

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

## Acknowledgements

The authors gratefully acknowledge the National Natural Science Foundation of China (22071104, C.L., 21871135, J.J., and 21871136, L.W.); Innovation support program of Jiangsu Province (BZ2023055, L.W.); The Starry Night Science Fund of Zhejiang University Shanghai Institute for Advanced Study (SN-ZJU-SIAS-006, L.W.).

## Author contributions

C.L. conceived the project. H.L. performed most of the experiments and wrote the manuscript. H.L. contributed theoretical calculations and single-crystal testing and resolution. Z.L. synthesized the macrocycle compound. C.L., J.J., and L.W. revised the manuscript. C.L., J.J., and L.W. supervised the project. All authors discussed and commented on the paper.

## Competing interests

The authors declare no competing interests.
