## [Peer Review File · Nature Communications]

Precise recognition of benzonitrile derivatives with supramolecular macrocycle of phosphorylated cavitand by co-crystallization methodREVIEWER COMMENTS

Reviewer #1 (Remarks to the Author):

The authors have chosen very rigid synthetic macrocyclic host of well-defined shallow cavity equipped with positively charged triangular region at the lower rim. The cavity is narrow and framed with preorganized hydrogen bond donors at the lower rim, while quite wide upper rim is surrounded with hydrogen bond acceptors (oxygen atoms of three phenylphosphine oxide groups).

The noteworthy result of the study is the finding that very precise structural and supramolecular match is required for the efficient formation of the host-guest inclusion complexes by this rigid macrocyclic host of preorganized hydrogen bond donors at the lower rim. The authors have provided an elegant supramolecular reasoning that only guests comprising benzonitrile fragment are suitable for the deep inclusion complexation and co-crystallization with this kind of macrocycle. Excitingly, of the several tested aromatic guests with partially negatively charged terminal groups, only benzonitrile successfully penetrates deeply into the cavity to reach and establish favorable hydrogen bonding with positively charged triangular lower rim. Also, this fundamental finding has been extended for the complexation of pharmaceutical molecules comprising benzonitrile fragments, namely, crisaborole and alectinib. These drug molecules have been successfully co-crystallized with the macrocyclic host, showing perspective that this macrocycle can act as a crystallization chaperone to help crystallize and structurally authenticate pharmaceutical molecules, that otherwise resist crystallization.

The macrocyclic host can also include either toluene, or fluorotoluene, or benzyl alcohol guests when crystallized from these solvents, correspondingly. However, the inclusion mode is shallow. In the case of dichloromethane, chlorobenzene and bromobenzene the solvent molecules reside at the periphery of the macrocycle, instead the cavity interacts with the aryl rings of adjacent macrocyclic molecules in the crystal.

I would argue with strictly applying the terminology of 'molecular docking' for the experimental co-crystallization studies of the host-guest complexes. There is a significant difference between computationally-based molecular docking and co-crystallization/ crystal structure determination. The molecular docking is computation/simulation screening that does not require the chemicals, solvents, time-consuming synthesis, crystallization and quite expensive single crystal X-ray diffraction experiments. Surely, host-guest co-crystallization studies and determination of the crystal structure of the obtained complexes to some extent mimic the molecular docking (and vice versa), as both methodologies rely on the molecular recognition, but purpose and resources of two approaches are entirely different. Please, comment on this.

Some points to consider:

When discussing crystal structure of the benzonitrile host-guest complex it is stated “In these interactions, the hydrogen atoms act as hydrogen bond donors, and the cyanide group acts as a hydrogen bond acceptor.” Actually, the donor of hydrogen bond is an atom that is covalently bound to hydrogen, not the hydrogen itself.

While discussing 2D Hirschfeld fingerprints of the same crystal structure it is stated : “The 2D Hirschfeld fingerprinting of F[3]A1-[P(O)Ph]3 and benzonitrile (Fig. 3i) further supports the importance of π - π stacking interactions (H...H and C-H...O).” However, neither H...H nor C-H...O contacts evidence the π - π stacking interactions. It is rather C...C contacts often referred to as π - π contacts, that are associated with the planar face-to-face stacking interactions. Also H...C contacts can mirror edge-to-face or edge-to-edge interactions, or CH... π interactions. Additionally, the provided fingerprints correspond to all contacts within the crystal space, not only of the guest contacts with cavity interior. Thus, other interactions contribute to these plots, for example, the interactions of macrocycle exterior with adjacent molecules in the crystal structure via π - π contacts and CH... π interactions.

In the Supplementary Information (page 3) it is stated: “The DELU command is used to make the atomic displacement parameters of the given atoms approximately isotropic”. This statement is incorrect. The DELU command restrains the ADPs of the atoms in the direction of the bond between them to be equal within the given standard uncertainty.

It appears from the CIF files that SQUEEZE procedure have been used to treat solvent molecules in many discussed structures, these should be clearly described in the Experimental Section. What was the reason to not include solvent molecules in the structure model? The SQUEEZE should not be applied routinely, the solvent molecules often have valuable contribution to the structure, interactions and stability of the crystals. Please, reconsider and provide structure model and refinement with solvent molecules included, where possible.

Some comments on crystal structures

Crystal structure G10_3-fluoro-4-hydroxybenzonitrile_F[3]A1-[P(O)Ph]3

The hydrogen atom H00T of the hydroxyl group of the guest molecule seems to be wrongly placed. The Alert level B is present in the checkcif file: PLAT420_ALERT_2_B D-H Bond Without Acceptor O00T --H00T. The authors have suggested an explanation that there is “no receptor for the oxygen atom on the hydroxyl group”. However, the close inspection of the close contacts in the crystal

structure model shows that this hydroxyl group is possibly engaged in the OH... π interaction with the phenyl ring of the adjacent macrocycle molecule, indeed O00T...C01P distance is of 3.12 Å suggesting weak OH... π hydrogen bond. Thus, hydrogen atom H00T should be placed along this interaction.

Crystal structure G1_benzonitrile_F[3]A1-[P(O)Ph]3

One of the guest benzonitrile molecules outside the macrocyclic cavity resides on the special position – an inversion center. The benzonitrile molecule does not fulfill the geometry of this symmetry element, since it is an example of the disorder about special position. Such type of disorder should be refined with the help of PART -1 instruction and account for the multiplicity of the special position - sof instruction 10.50 for atoms C6 and N5 are strongly recommended. Also, the residual density map tells us this. The chemical formula given in CIF file is incorrect 'C60 H39 O9 P3, 2(C7 H5 N), C4 H2 N' as there is no molecule C4 H2 N within the crystal complex. The proper chemical_formula_moiety is 'C60 H39 O9 P3, 2.5(C7 H5 N)'. The re-refinement and re-deposition of the structure model is recommended.

Crystal structure G5_4-fluoro-3-hydroxybenzonitrile_F[3]A1-[P(O)Ph]3

The hydrogen atom H00H of the hydroxyl group of the guest molecule seems to be wrongly placed. The Alert level C is present in the checkcif file: PLAT415_ALERT_2_C Short Inter D-H..H-X H00H ..H01J 2.00 Ang.. Also, inspection of the difference density map indicates misplaced hydrogen atom H00H, as a location (in red color around placed H00H atom) where there is no density to be modelled. The close inspection of the close contacts in the crystal structure model shows that this hydroxyl group is possibly engaged in the OH...O hydrogen bond with oxygen atom of the host, indeed O00H...O00G distance is of 3.26 Å suggesting weak OH...O hydrogen bond. Thus, hydrogen atom H00H should preferably be placed along this interaction.

Additionally, the difference density map suggests disorder of the Cl1 atom of the solvent molecule.

Crystal structure G2_3,4-difluorobenzonitrile_F[3]A1-[P(O)Ph]3

The inspection of the 'unsqueezed' data show that there is clearly additional solvent molecule dichloromethane C2 H4 Cl2 co-included into the macrocyclic cavity together with difluorobenzonitrile guest. The co-inclusion mode is probably similar as in the structure model G5_4-fluoro-3-hydroxybenzonitrile_F[3]A1-[P(O)Ph]3 comprising both guest and solvent molecule modelled inside the macrocyclic cavity. Why the solvent molecule have not been included into structure model during the refinement? The SQUEEZE should not be applied routinely, the solvent molecules often have valuable contribution to the structure, interactions and stability of the crystals. Please, reconsider and provide structure model and refinement with solvent molecules included.

Crystal structure toluene_F[3]A1-[P(O)Ph]3

One toluene molecule is introduced in the structure model as included into the cavity of the macrocycle. However, C01V...O00B distance of 2.68 Å between methyl carbon atom C01V of toluene and oxygen atom O00B of the macrocycle is too short for reasonable CH...O contact, while this distance is typical for OH...O hydrogen bond. Is it possible that actually it is not a toluene but phenol molecule included into the macrocycle and forming hydrogen bond with its hydroxyl group towards oxygen atom O00B of the host? Also C01V - C101W bond of 1.40 Ang is too short for C(sp3)-C(sp2) bond as indicated by Alert level C PLAT362_ALERT_2_C Short C(sp3)-C(sp2) Bond C01V - C01W . 1.40 Ang. Default C(sp3)-C(sp2) bond should be 1.52 Ang.

Oksana Danylyuk

Contour level: 0.4

- C
- H
- N
- O
- P
- Q

Contour level: 0.38

- C
- H
- Cl
- F
- N
- O
- P
- Q

Reviewer #2 (Remarks to the Author):

In this work, the authors reported a phenylphosphine oxide-bridged macrocycle, F[3]A1-[P(O)Ph]3, which could form co-crystallized host-guest complexes with various benzonitrile derivatives through non-covalent interactions. Moreover, two commercial drug molecules, crisaborole (anti-dermatitis) and alectinib (anti-cancer) with the benzonitrile fragment, could also form host-guest complexes with F[3]A1-[P(O)Ph]3 in the crystal state. This work may be publishable after addressing the following issues.

1. The phenylphosphine oxide-bridged macrocycle could form host-guest complexes with various benzonitrile derivatives in the solid state. The authors called them as key-lock complexes or “molecular docking”, which may be a trick. Are they really similar to the lock-and-key recognition or lock-key complexes of biological system? Why? The authors should provide more comments and explanations for these concepts.
2. The authors claimed the host-guest system in this work as “molecular docking” or key-lock complexation. In the process, how to define the “lock” state and “open” state? Can the lock and open processes be switchable? And how to achieve the lock and open processes?
3. During the host-guest or key-lock complexation, can any color and/or fluorescent changes be found? They may be interesting for their further practical applications.
4. The host-guest complexations in solution for all of the complexes are suggested to be studied, and one table for their binding constants can be provided.

Reviewer #3 (Remarks to the Author):

This is a submission with considerable merit, but also one that requires much work prior to publication. In essence the authors detail what appears to be a new phosphine oxide bridged rigid macrocycle and show it is quite selective in stabilizing crystalline complexes of substituted benzonitriles. The best result is obtaining a crystal structure of an alectinib, an FDA drug that according to the authors had never been subject to an X-ray diffraction analysis. This is a big deal and provides a small molecule complement to Makoto Fujita's award winning cage-based crystal structure determination approach. This is an aspect of the "story" that deserves more air time. It should drive acceptance of this work on its own. Conversely, the premise that the present crystallization approach can somehow inform or replace computational biological-substrate analyses strikes this reviewer as hype. Actually, a stronger word comes to mind, but it isn't appropriate for a professional document. Calculations are benchmarked to experiment and programs like Schrodinger are constantly recalibrating. There is almost no calibration with theory in the present submission. How do experiment and theory differ in their well-defined system. What are the errors in theory or computational structural analysis that their new receptor has unearthed?

The constant use of phrases like "will hold significant promise in simulating protein-targeted discovery methodologies" and "opens up possibilities for designing and optimizing novel drug candidates" have to be backed up with real results! Grandiose words like "perfect" and "remarkable" should also be purged. The authors should state their findings and express their opinions without inflating them through borrowed importance. The term pi-pi stacking, which refers only to geometry, should also be replaced by donor-acceptor, which constitutes a real interaction. The authors should also be honest in noting that there are thousands of host-guest complexes whose structure has been solved. So, only by relating their contribution to an actual unsolved problem and showing its solution can publication in Nature Communications be justified. The structure determination of drugs (as noted above) could provide this opportunity.

Responses to comments from Reviewer 1:

The authors have chosen very rigid synthetic macrocyclic host of well-defined shallow cavity equipped with positively charged triangular region at the lower rim. The cavity is narrow and framed with preorganized hydrogen bond donors at the lower rim, while quite wide upper rim is surrounded with hydrogen bond acceptors (oxygen atoms of three phenylphosphine oxide groups). The noteworthy result of the study is the finding that very precise structural and supramolecular match is required for the efficient formation of the host-guest inclusion complexes by this rigid macrocyclic host of preorganized hydrogen bond donors at the lower rim. The authors have provided an elegant supramolecular reasoning that only guests comprising benzonitrile fragment are suitable for the deep inclusion complexation and co-crystallization with this kind of macrocycle. Excitingly, of the several tested aromatic guests with partially negatively charged terminal groups, only benzonitrile successfully penetrates deeply into the cavity to reach and establish favorable hydrogen bonding with positively charged triangular lower rim. Also, this fundamental finding has been extended for the complexation of pharmaceutical molecules comprising benzonitrile fragments, namely, crisaborole and alectinib. These drug molecules have been successfully co-crystallized with the macrocyclic host, showing perspective that this macrocycle can act as a crystallization chaperone to help crystallize and structurally authenticate pharmaceutical molecules, that otherwise resist crystallization. The macrocyclic host can also include either toluene, or fluorotoluene, or benzyl alcohol guests when crystallized from these solvents, correspondingly. However, the inclusion mode is shallow. In the case of dichloromethane, chlorobenzene and bromobenzene the solvent molecules reside at the periphery of the macrocycle, instead the cavity interacts with the aryl rings of adjacent macrocyclic molecules in the crystal.

Response:

Thank you very much for your professional review of our manuscript. According to the suggestions, we have made corrections to our previous manuscript, and the detailed corrections are listed below.

Q1: I would argue with strictly applying the terminology of 'molecular docking' for the experimental co-crystallization studies of the host-guest complexes. There is a significant

difference between computationally-based molecular docking and co-crystallization/crystal structure determination. The molecular docking is computation/simulation screening that does not require the chemicals, solvents, time-consuming synthesis, crystallization and quite expensive single crystal X-ray diffraction experiments. Surely, host-guest co-crystallization studies and determination of the crystal structure of the obtained complexes to some extent mimic the molecular docking (and vice versa), as both methodologies rely on the molecular recognition, but purpose and resources of two approaches are entirely different. Please, comment on this.

Response:

Thanks a lot for your helpful comments and suggestions on our manuscript.

Our original idea is presented as follows: "Molecular docking" is a great technology for computer-aided drug design. The crucial phase in the molecular docking procedure involves the meticulous preparation of the structures of both the protein receptor and guest molecules, which are often obtained from sources such as X-ray crystallography or protein structure databases. (*Mol. Cancer*, **2016**,15, 64; *Nat. Biomed. Eng.*, **2022**, 6, 1180-1195; *J. Hematol. Oncol.*, **2020**,13,141; *Nat. Commun.* **2022**, 13, 6105.). Therefore, the molecular docking is closely related to the crystal structures. Just as reviewer 1 mentioned, in this work, our original idea was to mimic the molecular docking (and vice versa) to describe the specific binding of the host macrocyclic **F[3]A1-[P(O)Ph]₃** to benzonitrile derivatives based on co-crystallization experiments.

As reviewer 1 suggested, "molecular docking" may not be appropriately used for this work. In the revised manuscript, we changed "molecular docking" into "precise recognition" to describe the specific binding of host-guest co-crystallization studies and the title of the revised manuscript has been changed to: "Precise recognition of benzonitrile derivatives with supramolecular macrocycle of phosphorylated cavitand by co-crystallization method." Some parts of the manuscript have also been revised accordingly shown as follows:

1. We have revised the abstract section highlighted in yellow in the revised manuscript (page 1) as follows:

"Molecular docking serves to establish specific lock-and-key recognition between target proteins and drug molecules in the field of structure-based drug design. The importance of molecular docking in drug discovery lies in the precise recognition between potential drug compounds and their target receptors, which is generally based on the

computational method. However, it will become quite interesting if the rigid cavity structure of supramolecular macrocycles can precisely recognize a series of guests with specific fragments by mimicking molecular docking through co-crystallization experiments, instead of the conventional computational method."

2. We have revised the introduction section of the main text highlighted in yellow in the revised manuscript (page 3) as follows:

"However, the study of the precise recognition of a series of guests with specific fragments in the solid phase as the molecular docking does with supramolecular macrocycles is rare in supramolecular chemistry. Therefore, if the rigid cavity structure of supramolecular macrocycles can precisely recognize a series of guests with specific fragments by mimicking molecular docking through co-crystallization experiments, not only spatial and energetic complementarity with the target guest molecule can be achieved, but also the precision and reliability of this specific binding capability can be greatly improved. "

3. We modified "molecular docking" to "precise recognition" and "molecularly docked" to "precisely recognized", respectively, in the revised manuscript. (page 1, lines 4, 12, and 28; page 2, left, lines 1, and 3; page 2, right, lines 1, and 3; page 3, left, lines 1, 7, 11, 30, 32, and 37; page 4, left, lines 21; page 4, right, lines 8, and 10; page 5, right, lines 40, 41 and 50; page 6, left, line 11; page 6, right, lines 21, and 29; page 7, left, lines 2, and 4; page 8, left, lines 2, 12, and 18; page 8, right, line 9).

Q2: *When discussing crystal structure of the benzonitrile host-guest complex it is stated "In these interactions, the hydrogen atoms act as hydrogen bond donors, and the cyanide group acts as a hydrogen bond acceptor." Actually, the donor of hydrogen bond is an atom that is covalently bound to hydrogen, not the hydrogen itself.*

Response:

Thank you for your careful checks and we are sorry for the carelessness. According to your comments, and after reviewing the relevant literatures (*Nat. Chem.* **2023**, 15, 1559–1568; *J. Am. Chem. Soc.* **2017**, 139, 9325–9332; *J. Am. Chem. Soc.* **2016**, 138, 4334–4337), we have modified "the hydrogen atoms act as hydrogen bond donors" to "the phenyl **CH** act as a hydrogen bond donor " in the revised manuscript (page 5, left, line 4).

Q3: While discussing 2D Hirschfeld fingerprints of the same crystal structure it is stated : "The 2D Hirschfeld fingerprinting of $F[3]A1-[P(O)Ph]_3$ and benzonitrile (Fig. 3i) further supports the importance of π - π stacking interactions ($H\cdots H$ and $C-H\cdots O$)." However, neither $H\cdots H$ nor $C-H\cdots O$ contacts evidence the π - π stacking interactions. It is rather $C\cdots C$ contacts often referred to as π - π contacts, that are associated with the planar face-to-face stacking interactions. Also $H\cdots C$ contacts can mirror edge-to-face or edge-to-edge interactions, or $CH\cdots \pi$ interactions. Additionally, the provided fingerprints correspond to all contacts within the crystal space, not only of the guest contacts with cavity interior. Thus, other interactions contribute to these plots, for example, the interactions of macrocycle exterior with adjacent molecules in the crystal structure via π - π contacts and $CH\cdots \pi$ interactions.

Response:

Many thanks for the valuable comments and we are sorry for the confusion. According to your comments, we deleted the " π - π stacking interactions" and modified the sentence: "The **2D Hirschfeld** fingerprinting of **$F[3]A1-[P(O)Ph]_3$** and benzonitrile (**Fig. 3i**) further supports the importance of π - π stacking interactions (**$H\cdots H$** and **$C-H\cdots O$**) as the key driving forces for the formation of the key-lock complex. " into "The **2D Hirschfeld** fingerprinting of **$F[3]A1-[P(O)Ph]_3$** and benzonitrile (**Fig. 3i**) further supports the importance of donor-acceptor interactions (**$H\cdots H$** , **$C-H\cdots O$** , and **$C-H\cdots N$**) as the key driving forces for the formation of the key-lock complex. " in the revised manuscript (page 5, left, line 20).

Q4: In the Supplementary Information (page 3) it is stated: "The DELU command is used to make the atomic displacement parameters of the given atoms approximately isotropic". This statement is incorrect. The DELU command restrains the ADPs of the atoms in the direction of the bond between them to be equal within the given standard uncertainty.

Response:

We are thankful for the suggestion and we are sorry that this issue has arisen. We have modified the sentence: "The **DELU** command is used to make the atomic displacement parameters of the given atoms approximately isotropic" to "The **DELU** command restrains the **ADPs** of the atoms in the direction of the bond between them to

be equal within the given standard uncertainty. " in the revised Supplementary Information (page 3)

Q5: *It appears from the CIF files that SQUEEZE procedure have been used to treat solvent molecules in many discussed structures, these should be clearly described in the Experimental Section. What was the reason to not include solvent molecules in the structure model? The SQUEEZE should not be applied routinely, the solvent molecules often have valuable contribution to the structure, interactions and stability of the crystals. Please, reconsider and provide structure model and refinement with solvent molecules included, where possible.*

Response:

Many thanks for the valuable comments. According to your comments, we have re-refined the crystal data, and the resulting crystal structures are shown below. While the **SQUEEZE** procedure is not applied, after the crystal data of **F[3]A1-[P(O)Ph]₃ @ Guest** have been re-refined, the **CheckCIF** files for all crystal data contain no **Class A** and **B** alarms, except that the crystal structure (**F[3]A1-[P(O)Ph]₃ @ G15**) has some completely disordered dichloromethane solvent molecules and the **SQUEEZE** procedure was used to further refine the data. The reasons for using the **SQUEEZE** procedures were explained in the revised supporting information (page 3). All crystal data have been updated in the Cambridge Crystallographic Data Centre (**CCDC**). All crystal data tables in the support information have also been updated. The high-resolution crystal structures are shown below (**Figure R1-R23**).

Figure R1. Crystal structure of **F[3]A1-[P(O)Ph]₃ @ G1** (benzonitrile) (**CCDC: 2313229**)

After the crystal data of **F[31A1-[P(O)Ph]₃ @ G1** has been re-refined, the checkcif file for this crystal data does not contain any **Class A** and **B** alarms, while the **SQUEEZE** procedure is not applied.

Figure R2. Crystal structure of **F[31A1-[P(O)Ph]₃ @ G2** (3,4-difluorobenzonitrile) (CCDC: 2349291)

After the crystal data of **F[31A1-[P(O)Ph]₃ @ G2** has been re-refined, the checkcif file for this crystal data does not contain any **Class A** and **B** alarms, while the **SQUEEZE** procedure is not applied.

Figure R3. Crystal structure of **F[31A1-[P(O)Ph]₃ @ G3** (4-chlorobenzonitrile) (CCDC: 2349282)

After the crystal data of **F[31A1-[P(O)Ph]₃ @ G3** has been re-refined, the checkcif file for this crystal data does not contain any **Class A** and **B** alarms, while the **SQUEEZE** procedure is not applied.

Figure R4. Crystal structure of **F[3]A1-[P(O)Ph]₃ @ G4** (4-bromobenzonitrile) (CCDC: 2349283)

After the crystal data of **F[3]A1-[P(O)Ph]₃ @ G4** has been re-refined, the checkcif file for this crystal data does not contain any **Class A** and **B** alarms, while the **SQUEEZE** procedure is not applied.

Figure R5. Crystal structure of **F[3]A1-[P(O)Ph]₃ @ G5** (4-fluoro-3-hydroxybenzonitrile) (CCDC: 2349290)

After the crystal data of **F[3]A1-[P(O)Ph]₃ @ G5** has been re-refined, the checkcif file for this crystal data does not contain any **Class A** and **B** alarms, while the **SQUEEZE** procedure is not applied.

Figure R6. Crystal structure of **F[3]A1-[P(O)Ph]₃ @ G6** (4-trifluoromethylbenzonitrile) (**CCDC: 2349280**)

After the crystal data of **F[3]A1-[P(O)Ph]₃ @ G6** has been re-refined, the checkcif file for this crystal data does not contain any **Class A** and **B** alarms, while the **SQUEEZE** procedure is not applied.

Figure R7. Crystal structure of **F[3]A1-[P(O)Ph]₃ @ G7** (4-ethylbenzonitrile) (**CCDC: 2349292**)

After the crystal data of **F[3]A1-[P(O)Ph]₃ @ G7** has been re-refined, the checkcif file for this crystal data does not contain any **Class A** and **B** alarms, while the **SQUEEZE** procedure is not applied.

Figure R8. Crystal structure of **F[3]A1-[P(O)Ph]₃ @ G8** (4-butylbenzotrile) (CCDC: 2313234)

After the crystal data of **F[3]A1-[P(O)Ph]₃ @ G8** has been re-refined, the checkcif file for this crystal data does not contain any **Class A** and **B** alarms, while the **SQUEEZE** procedure is not applied.

Figure R9. Crystal structure of **F[3]A1-[P(O)Ph]₃ @ G9** (3,4-dihydroxybenzotrile) (CCDC: 2349288)

After the crystal data of **F[3]A1-[P(O)Ph]₃ @ G9** has been re-refined, the checkcif file for this crystal data does not contain any **Class A** and **B** alarms, while the **SQUEEZE** procedure is not applied.

Figure R10. Crystal structure of **F[3]A1-[P(O)Ph]₃ @ G10** (3-fluoro-4-hydroxybenzonitrile)
(CCDC: 2313225)

After the crystal data of **F[3]A1-[P(O)Ph]₃ @ G10** has been re-refined, the checkcif file for this crystal data does not contain any **Class A** and **B** alarms, while the **SQUEEZE** procedure is not applied.

Figure R11. Crystal structure of **F[3]A1-[P(O)Ph]₃ @ G11** (4-cyanobenzenesulfonamide)
(CCDC: 2349285)

After the crystal data of **F[3]A1-[P(O)Ph]₃ @ G11** has been re-refined, the checkcif file for this crystal data does not contain any **Class A** and **B** alarms, while the **SQUEEZE** procedure is not applied.

Figure R12. Crystal structure of **F[31A1-[P(O)Ph]₃ @ G12** (4-Hydroxy-4-biphenylcarbonitrile)
(CCDC: 2313221)

After the crystal data of **F[31A1-[P(O)Ph]₃ @ G12** has been re-refined, the checkcif file for this crystal data does not contain any **Class A** and **B** alarms, while the **SQUEEZE** procedure is not applied.

Figure R13. Crystal structure of **F[31A1-[P(O)Ph]₃ @ G13** (1,4-bis(4-cyanostyryl)benzene)
(CCDC: 2349287)

After the crystal data of **F[31A1-[P(O)Ph]₃ @ G13** has been re-refined, the checkcif file for this crystal data does not contain any **Class A** and **B** alarms, while the **SQUEEZE** procedure is not applied.

Figure R14. Crystal structure of **F[31A1-[P(O)Ph]₃ @ G14** (crisaborole) (**CCDC: 2313217**)

After the crystal data of **F[31A1-[P(O)Ph]₃ @ G14** has been re-refined, the checkcif file for this crystal data does not contain any **Class A** and **B** alarms, while the **SQUEEZE** procedure is not applied.

Figure R15. Crystal structure of **F[31A1-[P(O)Ph]₃ @ G15** (alectinib) (**CCDC: 2313233**)

Following your suggestion, the crystal data of **F[31A1-[P(O)Ph]₃ @ G15** has been re-refined. We have tried to contain as many solvent molecules as possible in the crystal structure as you suggested, but there are still some completely disordered dichloromethane solvent molecules, so the **SQUEEZE** procedure was used to further refine the crystal data.

Figure R16. Crystal structure of **F[3]A1-[P(O)Ph]₃** @ 3-phenylpropanonitrile (**CCDC: 2313219**)

After the crystal data of **F[3]A1-[P(O)Ph]₃** @ 3-phenylpropanonitrile has been re-refined, the checkcif file for this crystal data does not contain any **Class A** and **B** alarms, while the **SQUEEZE** procedure is not applied.

Figure R17. Crystal structure of **F[3]A1-[P(O)Ph]₃** @ benzylalcohol (**CCDC: 2313236**)

After the crystal data of **F[3]A1-[P(O)Ph]₃** @ benzylalcohol has been re-refined, the checkcif file for this crystal data does not contain any **Class A** and **B** alarms, while the **SQUEEZE** procedure is not applied.

Figure R18. Crystal structure of **F[3]A1-[P(O)Ph]₃** @ chlorobenzene (CCDC: 2313220)

After the crystal data of **F[3]A1-[P(O)Ph]₃** @ chlorobenzene has been re-refined, the checkcif file for this crystal data does not contain any **Class A** and **B** alarms, while the **SQUEEZE** procedure is not applied.

Figure R19. Crystal structure of **F[3]A1-[P(O)Ph]₃** @ bromobenzene (CCDC: 2313216)

After the crystal data of **F[3]A1-[P(O)Ph]₃** @ bromobenzene has been re-refined, the checkcif file for this crystal data does not contain any **Class A** and **B** alarms, while the **SQUEEZE** procedure is not applied.

Figure R20. Crystal structure of **F[3]A1-[P(O)Ph]₃** @ toluene (CCDC: 2349293)

After the crystal data of **F[3]A1-[P(O)Ph]₃** @ toluene has been re-refined, the checkcif file for this crystal data does not contain any **Class A** and **B** alarms, while the **SQUEEZE** procedure is not applied.

Figure R21. Crystal structure of **F[3]A1-[P(O)Ph]₃** @ 4-hydroxybenzylcyanide (CCDC: 2349281)

After the crystal data of **F[3]A1-[P(O)Ph]₃** @ 4-hydroxybenzylcyanide has been re-refined, the checkcif file for this crystal data does not contain any **Class A** and **B** alarms, while the **SQUEEZE** procedure is not applied.

Figure R22. Crystal structure of $F[3]A1-[P(O)Ph]_3$ @ p-fluorotoluene (CCDC: 2313215)

After the crystal data of $F[3]A1-[P(O)Ph]_3$ @ p-fluorotoluene has been re-refined, the checkcif file for this crystal data does not contain any **Class A** and **B** alarms, while the **SQUEEZE** procedure is not applied.

Figure R23. Crystal structure of $F[3]A1-[P(O)Ph]_3$ @ dichloromethane (CCDC: 2349284)

After the crystal data of $F[3]A1-[P(O)Ph]_3$ @ dichloromethane has been re-refined, the checkcif file for this crystal data does not contain any **Class A** and **B** alarms, while the **SQUEEZE** procedure is not applied.

Q6 : Some comments on crystal structures. Crystal structure G10_3-fluoro-4-hydroxybenzotrile_F[3]A1-[P(O)Ph]₃. The hydrogen atom H00T of the hydroxyl group of the guest molecule seems to be wrongly placed. The Alert level B is present in the checkcif file: PLAT420_ALERT_2_B D-H Bond Without Acceptor O00T --H00T. The authors have suggested an explanation that there is “no receptor for the oxygen atom on the hydroxyl group”. However, the close inspection of the close contacts in the crystal structure model shows that this hydroxyl group is possibly engaged in the OH..pi interaction with the phenyl ring of the adjacent macrocycle molecule, indeed O00T..C01P distance is of 3.12 Å suggesting weak OH..pi hydrogen bond. Thus, hydrogen atom H00T should be placed along this interaction.

Response:

Thank you very much for the profound suggestions. According to your suggestion, we carefully observed the close contact of the guest molecule **G10** (3-fluoro-4-hydroxybenzotrile) with the adjacent **F[3]A1-[P(O)Ph]₃** molecule and, as you stated, the distance between the **O00T** on the **G10** molecule and the **C01P** of the adjacent **F[3]A1-[P(O)Ph]₃** molecule is 3.196 Å (**Figure R24**).

Figure R24. The crystalline structure of **F[3]A1-[P(O)Ph]₃ @G10**. Solvents have been removed for clarity.

Subsequently, we placed the hydrogen atom **H00T** along this interaction and fixed the hydrogen atom **H00T** (**Figure R25a**). After several refinements, the Checkcif file still had a level **B** Alert: **PLAT420 ALERT 2_B D-H Bond Without Acceptor O00T --H00T**, which was probably caused by the large distance between the **O00T** and the **C01P**. Alternatively, we attempted to place the hydrogen atoms **H00T** along **O00F** and **O00T** (**Figure R25b**), and the **Alert** level **B** was removed. We uploaded the updated crystal data to the **CCDC** crystal database (**CCDC: 2313225**).

Figure R25. The crystalline structure of **F[3]A1-[P(O)Ph]₃ @G10**. Solvents have been removed for clarity.

Q7:Crystal structure *G1_benzonitrile_F[3]A1-[P(O)Ph]₃*. One of the guest benzonitrile molecules outside the macrocyclic cavity resides on the special position – an inversion center. The benzonitrile molecule does not fulfill the geometry of this symmetry element, since it is an example of the disorder about special position. Such type of disorder should be refined with the help of *PART -1* instruction and account for the multiplicity of the special position - *sof* instruction 10.50 for atoms *C6* and *N5* are strongly recommended. Also, the residual density map tells us this. The chemical formula given in CIF file is incorrect '*C60 H39 O9 P3, 2(C7 H5 N), C4 H2 N*' as there is no molecule *C4 H2 N* within the crystal complex. The proper chemical_formula_moiety is '*C60 H39 O9 P3, 2.5(C7 H5 N)*'. The re-refinement and re-deposition of the structure model is recommended.

Response:

Thanks for the constructive suggestions. In accordance with your comments, we carefully examined the crystal data **G1_benzonitrile_F[3]A1-[P(O)Ph]₃** and found a guest benzonitrile molecule outside the macrocyclic cavity in a special position (**Figure R26a-**

b). The benzonitrile molecule was refined using the 'PART -1 -c' and 'PART -1 10.50' instructions (**Figure R26c**), resulting in the proper chemical_formula_moiety is 'C6O H39 O9 P3, 2.5(C7 H5 N)'. We uploaded the updated crystal data to the **CCDC** crystal database (CCDC: 2313229).

Figure R26. The crystalline structure of **F[3]A1-[P(O)Ph]₃ @ G1**. Solvents have been removed for clarity.

Q8: Crystal structure *G5_4-fluoro-3-hydroxybenzonitrile_F[3]A1-[P(O)Ph]₃*. The hydrogen atom **H00H** of the hydroxyl group of the guest molecule seems to be wrongly placed. The Alert level C is present in the checkcif file: *PLAT415_ALERT_2_C Short Inter D-H..H-X H00H..H01J 2.00 Ang..* Also, inspection of the difference density map indicates misplaced hydrogen atom **H00H**, as a location (in red color around placed **H00H** atom) where there is no density to be modelled. The close inspection of the close contacts in the crystal structure model shows that this hydroxyl group is possibly engaged in the *OH..O* hydrogen bond with oxygen atom of the host, indeed *O00H..O00G* distance is of 3.26 Å suggesting weak *OH..O* hydrogen bond. Thus, hydrogen atom **H00H** should preferably be placed along this interaction. Additionally, the difference density map suggests disorder of the **C11** atom of the solvent molecule.

Response:

Thanks for the good suggestion. According to your suggestion, the crystal data *G5_4-fluoro-3-hydroxybenzonitrile_F[3]A1-[P(O)Ph]₃* were scrutinized and it was found that there was indeed a problem with the position of the hydrogen atom **H00H** on the hydroxyl group of the guest molecule. Therefore, the hydrogen atom **H00H** was then placed along

the **O00H...O00G** interactions and as a result **PLAT415 ALERT 2_C** was removed (**Figure R27**). In addition, the **Cl1** atom in the solvent molecule in a disordered state has been solved using the **'PART'** instruction. We uploaded the updated crystal data to the **CCDC** crystal database (CCDC: 2349290).

Figure R27. The crystalline structure of **F[3]A1-[P(O)Ph]₃ @ G5**. Solvents have been removed for clarity.

Q9: *Crystal structure G2_3,4-difluorobenzonitrile_F[3]A1-[P(O)Ph]₃. The inspection of the 'unsqueezed' data show that there is clearly additional solvent molecule dichloromethane C2 H4 Cl2 co-included into the macrocyclic cavity together with difluorobenzonitrile guest. The co-inclusion mode is probably similar as in the structure model G5_4-fluoro-3-hydroxybenzonitrile_F[3]A1-[P(O)Ph]₃ comprising both guest and solvent molecule modelled inside the macrocyclic cavity. Why the solvent molecule have not been included into structure model during the refinement? The SQUEEZE should not be applied routinely, the solvent molecules often have valuable contribution to the structure, interactions and stability of the crystals. Please, reconsider and provide structure model and refinement with solvent molecules included.*

Response:

Thank you for your kind suggestion, and we are sorry for this problem. Following your suggestion, after the crystal data of **F[3]A1-[P(O)Ph]₃ @ 4-hydroxybenzylcyanide** has been re-refined, the checkcif file for this crystal data does not contain any **Class A** and **B** alarms,

while the **SQUEEZE** procedure is not applied. And the high-resolution crystal structure is shown below (**Figure R28**). We uploaded the updated crystal data to the **CCDC** crystal database (CCDC: 2349291).

Figure R28. The crystalline structure of **F[3]A1-[P(O)Ph]₃ @ G2** (CCDC: 2349291)

Q10 Crystal structure toluene_F[3]A1-[P(O)Ph]₃. One toluene molecule is introduced in the structure model as included into the cavity of the macrocycle. However, C01V...O00B distance of 2.68 Å between methyl carbon atom C01V of toluene and oxygen atom O00B of the macrocycle is too short for reasonable CH...O contact, while this distance is typical for OH...O hydrogen bond. Is it possible that actually it is not a toluene but phenol molecule included into the macrocycle and forming hydrogen bond with its hydroxyl group towards oxygen atom O00B of the host? Also C01V - C101W bond of 1.40 Ang is too short for C(sp³)-C(sp²) bond as indicated by Alert level C PLAT362_ALERT_2_C Short C(sp³)-C(sp²) Bond C01V - C01W . 1.40 Ang. Default C(sp³)-C(sp²) bond should be 1.52 Ang.

Response:

Many thanks for the valuable comments. We have examined these crystal data and, as you pointed out, the length of the **C01W-C01V** bond is 1.40 Å (**Figure R29a**). If **C01V** is referred to as the oxygen atom **O01V**, the temperature factor (U_{eq}) of the oxygen atom model becomes larger compared to the carbon atom after several refinements (**Figure R29b**), which also results in a worse **R₁** value. In addition, we carefully examined the experimental record, which showed that this co-crystal was obtained by slowly evaporating a saturated toluene solution of **F[3]A1-[P(O)Ph]₃**, where no phenol was used.

Figure R29. The crystalline structure of **F[3]A1-[P(O)Ph]₃ @ toluene.**

Figure R30. The crystalline structure of **a.) Pillar[n]arenes encapsulated *o*-xylene;**
b.) Pillar[n]arenes encapsulated *p*-xylene

In order to clarify this issue, we reviewed the literatures. In crystal data of Pillar[n]arenes encapsulated *o*-xylene (**Figure R30a**) (*J. Am. Chem. Soc.* **2018**, 140, 6921-6930; **CCDC**: 1817587), it is shown that authors used the restriction command **DFIX 1.49 C42 C43** for bond **C42-C43**, resulting in a bond length of 1.471 Å. When this constraint is removed, the length of the **C42-C43** bond is 1.355 Å. This could be attributed to the fact that *o*-xylene is affected by multiple forces in the cavity, leading to a contraction

of the **C(sp³)-C(sp²)** bond to adapt to its environment. Similarly, this situation is also presented in the crystal structure in **Figure R30b**, when some constraint is removed, the bond length of bond **C36-C37** is 1.24 Å. (*Angew. Chem. Int. Ed.* **2018**, 57, 12845-12849).

Based on the above literature discussion and study, in the crystal structure of toluene_F[3]A1-[P(O)Ph]₃, after the **C01W-C01V** bond is used with the restriction instruction "**DFIX 1.5 0.01 C01W C01V**", the length of the **C01W-C01V** bond is 1.45 Å, and therefore, **Alert level C PLAT362_ALERT_2_C** was removed. We uploaded the updated crystal data to the **CCDC** crystal database (CCDC: 2349293).

Responses to comments from Reviewer 2:

In this work, the authors reported a phenylphosphine oxide-bridged macrocycle, F[3]A1-[P(O)Ph]3, which could form co-crystallized host-guest complexes with various benzonitrile derivatives through non-covalent interactions. Moreover, two commercial drug molecules, crisaborole (anti-dermatitis) and alectinib (anti-cancer) with the benzonitrile fragment, could also form host-guest complexes with F[3]A1-[P(O)Ph]3 in the crystal state. This work may be publishable after addressing the following issues.

Response:

We greatly appreciate your positive comments and helpful suggestions on improving the quality of our manuscript. These comments are all valuable and helpful for improving our work. According to your comments, we have made changes to our manuscript and added additional data to make our results convincing. In addition, all of your concerns have been addressed in detail as follows, and the corresponding changes have also been made in the revised manuscript.

Q1: *The phenylphosphine oxide-bridged macrocycle could form host-guest complexes with various benzonitrile derivatives in the solid state. The authors called them as key-lock complexes or “molecular docking”, which may be a trick. Are they really similar to the lock-and-key recognition or lock-key complexes of biological system? Why? The authors should provide more comments and explanations for these concepts.*

Response:

Thank you very much for the good questions. The term "key-lock complexes" comes from the biological "key-lock model" (*Nature*, **2010**, 464, 575–578; *Nat. Chem. Biol.* **2021**, 17, 1214-1216.), which states that the structures of the enzyme and substrate matched at the binding site and explains the specific recognition between enzyme and substrate. In this work, after the screening of various guest molecules containing benzene rings with partially negatively charged end groups, a significant and good binding affinity was observed between the partially negatively charged **-CN** end groups and the host macrocycle **F[3]A1-[P(O)Ph]3**, in which the hydrogen atom at the lower rim of the supramolecular macrocycle **F[3]A1-[P(O)Ph]3** is partially positively charged. This interaction showed structural complementarity similar to that of a lock and key model.

In terms of the term "Molecular docking", our original idea is presented as follows: "Molecular docking" is a great technology for computer-aided drug design. The crucial phase in the molecular docking procedure involves the meticulous preparation of the structures of both the protein receptor and guest molecules, which are often obtained from sources such as X-ray crystallography or protein structure databases. (*Mol. Cancer*, **2016**, *15*, 64; *Nat. Biomed. Eng.*, **2022**, *6*, 1180-1195; *J. Hematol. Oncol.*, **2020**, *13*, 141; *Nat. Commun.* **2022**, *13*, 6105.). Therefore, the molecular docking is closely related to the crystal structures. Just as reviewer 1 mentioned, in this work, our original idea was to mimic the molecular docking (and vice versa) to describe the specific binding of the host macrocyclic **F[3]A1-[P(O)Ph]₃** to benzonitrile derivatives based on co-crystallization experiments.

As reviewers suggested, "molecular docking" may not be appropriately used for this work. In the revised manuscript, we changed "molecular docking" into "precise recognition" to describe the specific binding of host-guest co-crystallization studies and the title of the revised manuscript has been changed to: "*Precise recognition of benzonitrile derivatives with supramolecular macrocycle of phosphorylated cavitanol by co-crystallization method.*" Some parts of the manuscript have also been revised accordingly shown as follows:

1. We have revised the abstract section highlighted in yellow in the revised manuscript (page 1) as follows:

"Molecular docking serves to establish specific lock-and-key recognition between target proteins and drug molecules in the field of structure-based drug design. The importance of molecular docking in drug discovery lies in the precise recognition between potential drug compounds and their target receptors, which is generally based on the computational method. However, it will become quite interesting if the rigid cavity structure of supramolecular macrocycles can precisely recognize a series of guests with specific fragments by mimicking molecular docking through co-crystallization experiments, instead of the conventional computational method."

2. We have revised the introduction section of the main text highlighted in yellow in the revised manuscript (page 3) as follows:

"However, the study of the precise recognition of a series of guests with specific fragments in the solid phase as the molecular docking does with supramolecular macrocycles is rare in supramolecular chemistry. Therefore, if the rigid cavity structure of supramolecular macrocycles can precisely recognize a series of guests with specific

fragments by mimicking molecular docking through co-crystallization experiments, not only spatial and energetic complementarity with the target guest molecule can be achieved, but also the precision and reliability of this specific binding capability can be greatly improved.

3. We modified "molecular docking" to "precise recognition" and "molecularly docked" to "precisely recognized", respectively, in the revised manuscript. (page 1, lines 4, 12, and 28; page 2, left, lines 1, and 3; page 2, right, lines 1, and 3; page 3, left, lines 1, 7, 11, 30, 32, and 37; page 4, left, lines 21; page 4, right, lines 8, and 10; page 5, right, lines 40, 41 and 50; page 6, left, line 11; page 6, right, line 21, and 29; page 7, left, line 2, and 4; page 8, left, line 2, 12, and 18; page 8, right, line 9).

Q2: . The authors claimed the host-guest system in this work as “molecular docking” or key-lock complexation. In the process, how to define the “lock” state and “open” state? Can the lock and open processes be switchable? And how to achieve the lock and open processes?

Response:

Thank you for your professional suggestions. The term "key-lock complexes" comes from the biological "key-lock model" (*Nature*, **2010**, 464, 575–578; *Nat. Chem. Biol.* **2021**, 17, 1214-1216.), which states that the structures of the enzyme and substrate matched at the binding site. The key-lock model does not necessarily have a "locked" state and an "open" state. In this work, the supramolecular macrocycle **F[3]A1-[P(O)Ph]₃** showed stable interactions with benzonitrile and its derivatives, similar to the structural complementarity described in the "key and lock" model.

Q3: During the host-guest or key-lock complexation, can any color and/or fluorescent changes be found? They may be interesting for their further practical applications.

Response:

Thank you for your constructive comments. No significant color or fluorescence changes were observed during the study of the guest molecules and the host macrocycle **F[3]A1-[P(O)Ph]₃**.

Q4: The host-guest complexations in solution for all of the complexes are suggested to be studied, and one table for their binding constants can be provided.

Response:

Thank you for your valuable comments. According to your suggestions, we have measured and calculated the binding constants of all host-guest complexes that appear in the manuscript by UV-vis titration experiments, and the details are summarized in **Table R1**. Details of UV-vis titration experiments of all host-guest complexes are shown in **Figure R32-R46**. A corresponding discussion has also been added in the revised manuscript: "Binding constants between **F[3]A1-[P(O)Ph]₃** and other guests were also determined (**Fig. S39-S53**). "(page 5, right, line 36) in the revised manuscript. And, all supplementary experimental details have been added to the supporting information (**Fig. S39-S53**).

Table R1. Details of binding constants for all host-guest complexes formed

Host-guest complexes	K_a	R^2
F[3]A1-[P(O)Ph] ₃ @ G1 (benzonitrile)	$(4.648 \pm 0.372) \times 10^3 \text{ M}^{-1}$	> 0.998
F[3]A1-[P(O)Ph] ₃ @ G2 (3,4-difluorobenzonitrile)	$(1.087 \pm 0.068) \times 10^4 \text{ M}^{-1}$	> 0.991
F[3]A1-[P(O)Ph] ₃ @ G3 (4-chlorobenzonitrile)	$(1.865 \pm 0.087) \times 10^4 \text{ M}^{-1}$	> 0.993
F[3]A1-[P(O)Ph] ₃ @ G4 (4-bromobenzonitrile)	$(1.972 \pm 0.046) \times 10^4 \text{ M}^{-1}$	> 0.998
F[3]A1-[P(O)Ph] ₃ @ G5 (4-fluoro-3-hydroxybenzonitrile)	$(6.754 \pm 0.356) \times 10^3 \text{ M}^{-1}$	> 0.993
F[3]A1-[P(O)Ph] ₃ @ G6 (4-trifluoromethylbenzonitrile)	$(6.253 \pm 0.377) \times 10^3 \text{ M}^{-1}$	> 0.994
F[3]A1-[P(O)Ph] ₃ @ G7 (4-ethylbenzonitrile)	$(6.446 \pm 0.339) \times 10^3 \text{ M}^{-1}$	> 0.993
F[3]A1-[P(O)Ph] ₃ @ G8 (4-butylbenzonitrile)	$(8.065 \pm 0.491) \times 10^3 \text{ M}^{-1}$	> 0.991
F[3]A1-[P(O)Ph] ₃ @ G9 (3,4-dihydroxybenzonitrile)	$(1.808 \pm 0.041) \times 10^4 \text{ M}^{-1}$	> 0.998
F[3]A1-[P(O)Ph] ₃ @ G10 (3-fluoro-4-hydroxybenzonitrile)	$(1.455 \pm 0.051) \times 10^4 \text{ M}^{-1}$	> 0.996
F[3]A1-[P(O)Ph] ₃ @ G11 (4-cyanobenzenesulfonamide)	$(1.344 \pm 0.037) \times 10^4 \text{ M}^{-1}$	> 0.998
F[3]A1-[P(O)Ph] ₃ @ G12 (4'-Hydroxy-4-biphenylcarbonitrile)	$(1.932 \pm 0.055) \times 10^4 \text{ M}^{-1}$	> 0.997
F[3]A1-[P(O)Ph] ₃ @ G13 (1,4-bis(4-cyanostyryl)benzene)	$(2.158 \pm 0.076) \times 10^4 \text{ M}^{-1}$	> 0.996
F[3]A1-[P(O)Ph] ₃ @ G14 (crisaborole)	$(1.150 \pm 0.063) \times 10^4 \text{ M}^{-1}$	> 0.991
F[3]A1-[P(O)Ph] ₃ @ G15 (alelectinib)	$(1.669 \pm 0.046) \times 10^4 \text{ M}^{-1}$	> 0.997

To determine the association constant for the complexation between **F[3]A1-[P(O)Ph]₃** and the guests (**G1-G15**), UV-vis titration experiments were carried out in **CHCl₃**, which had a constant concentration of **F[3]A1-[P(O)Ph]₃** (0.01 mM) and varying concentrations of the guests (**G1-G15**). By a non-linear curve-fitting method, the association constant (**K_a**) of **F[3]A1-[P(O)Ph]₃** @ the guests (**G1-G15**) were estimated. The non-linear curve fittings were based on the following equation:

$$\Delta A = (\Delta A_{\infty} / [H]_0) * (0.5 * [G]_0 + 0.5 * ([H]_0 + 1 / K_a) - (0.5 * ([G]_0^2 + (2 * [G]_0 * (1 / K_a - [H]_0)) + (1 / K_a + [H]_0)^2)^{0.5}))$$

Where the ΔA is the UV-vis absorption change of **F[3]A1-[P(O)Ph]₃** upon addition of the guests (**G1-G15**), ΔA_{∞} is the UV-vis absorption changes at 265 nm when the **F[3]A1-[P(O)Ph]₃** is completely complexed, $[H]_0$ is the initial concentration of **F[3]A1-[P(O)Ph]₃**, and $[G]_0$ is the varying concentration of the guests (**G1-G15**).

Figure R31. (a). UV-Vis absorption spectra of **F[31A1-[P(O)Ph]₃** at a constant concentration of 0.01 mM with different concentrations of **G1** (benzonitrile), ranging from 0 mM to 1.38 mM. (b). The absorbance changes of **F[31A1-[P(O)Ph]₃** upon the addition of **G1**. The red solid line was obtained from the non-linear curve fitting. The association constant (K_a) between **F[31A1-[P(O)Ph]₃** and **G1** was estimated to be $(4.648 \pm 0.372) \times 10^3 \text{ M}^{-1}$. (c). UV-Vis absorption spectra of complex **F[31A1-[P(O)Ph]₃ @ G1** with different molar ratios in water while $[\text{F[31A1-[P(O)Ph]}_3] + [\text{G1}] = 0.10 \text{ mM}$. (d). Job plots of the complex **F[31A1-[P(O)Ph]₃ @ G1** showing a 1:1 stoichiometry between **F[31A1-[P(O)Ph]₃** and **G1** by plotting the absorbance differences at 265 nm (a characteristic absorption peak of **F[31A1-[P(O)Ph]₃**) against the mole fraction of **G1**.

Figure R32. (a). UV-Vis absorption spectra of $F[31A1-[P(O)Ph]_3$ at a constant concentration of 0.01 mM with different concentrations of **G2** (3,4-difluorobenzonitrile), ranging from 0 mM to 0.74 mM. (b). The absorbance changes of $F[31A1-[P(O)Ph]_3$ upon the addition of **G2**. The red solid line was obtained from the non-linear curve fitting. The association constant (K_a) between $F[31A1-[P(O)Ph]_3$ and **G2** was estimated to be $(1.0876 \pm 0.0689) \times 10^4 \text{ M}^{-1}$. (c). UV-Vis absorption spectra of complex $F[31A1-[P(O)Ph]_3 @ G2$ with different molar ratios in water while $[F[31A1-[P(O)Ph]_3] + [G2] = 0.10 \text{ mM}$. (d). Job plots of the complex $F[31A1-[P(O)Ph]_3 @ G2$ showing a 1:1 stoichiometry between $F[31A1-[P(O)Ph]_3$ and **G2** by plotting the absorbance differences at 265 nm (a characteristic absorption peak of $F[31A1-[P(O)Ph]_3$) against the mole fraction of **G2**.

Figure R33. (a). UV-Vis absorption spectra of $F[31A1-[P(O)Ph]_3$ at a constant concentration of 0.01 mM with different concentrations of $G3$ (4-chlorobenzonitrile), ranging from 0 mM to 0.51 mM. (b). The absorbance changes of $F[31A1-[P(O)Ph]_3$ upon the addition of $G3$. The red solid line was obtained from the non-linear curve fitting. The association constant (K_a) between $F[31A1-[P(O)Ph]_3$ and $G3$ was estimated to be $(1.8652 \pm 0.0877) \times 10^4 M^{-1}$. (c). UV-Vis absorption spectra of complex $F[31A1-[P(O)Ph]_3 @ G3$ with different molar ratios in water while $[F[31A1-[P(O)Ph]_3] + [G3] = 0.10$ mM. (d). Job plots of the complex $F[31A1-[P(O)Ph]_3 @ G3$ showing a 1:1 stoichiometry between $F[31A1-[P(O)Ph]_3$ and $G3$ by plotting the absorbance differences at 265 nm (a characteristic absorption peak of $F[31A1-[P(O)Ph]_3$) against the mole fraction of $G3$.

Figure R34. (a). UV-Vis absorption spectra of $F[31A1]-[P(O)Ph]_3$ at a constant concentration of 0.01 mM with different concentrations of **G4** (4-bromobenzonitrile), ranging from 0 mM to 0.57 mM. (b). The absorbance changes of $F[31A1]-[P(O)Ph]_3$ upon the addition of **G4**. The red solid line was obtained from the non-linear curve fitting. The association constant (K_a) between $F[31A1]-[P(O)Ph]_3$ and **G4** was estimated to be $(1.9722 \pm 0.0462) \times 10^4 \text{ M}^{-1}$. (c). UV-Vis absorption spectra of complex $F[31A1]-[P(O)Ph]_3 @ \text{G4}$ with different molar ratios in water while $[F[31A1]-[P(O)Ph]_3] + [\text{G4}] = 0.10 \text{ mM}$. (d). Job plots of the complex $F[31A1]-[P(O)Ph]_3 @ \text{G4}$ showing a 1:1 stoichiometry between $F[31A1]-[P(O)Ph]_3$ and **G4** by plotting the absorbance differences at 265 nm (a characteristic absorption peak of $F[31A1]-[P(O)Ph]_3$) against the mole fraction of **G4**.

Figure R35. (a). UV-Vis absorption spectra of F[31A1-[P(O)Ph]₃ at a constant concentration of 0.01 mM with different concentrations of G5 (4-fluoro-3-hydroxybenzonitrile), ranging from 0 mM to 0.54 mM. (b). The absorbance changes of F[31A1-[P(O)Ph]₃ upon the addition of G5. The red solid line was obtained from the non-linear curve fitting. The association constant (K_a) between F[31A1-[P(O)Ph]₃ and G5 was estimated to be $(6.754 \pm 0.356) \times 10^3 \text{ M}^{-1}$. (c). UV-Vis absorption spectra of complex F[31A1-[P(O)Ph]₃ @ G5 with different molar ratios in water while [F[31A1-[P(O)Ph]₃] + [G5] = 0.10 mM. (d). Job plots of the complex F[31A1-[P(O)Ph]₃ @ G5 showing a 1:1 stoichiometry between F[31A1-[P(O)Ph]₃ and G5 by plotting the absorbance differences at 265 nm (a characteristic absorption peak of F[31A1-[P(O)Ph]₃) against the mole fraction of G5.

Figure R36. (a). UV-Vis absorption spectra of F[31A1-[P(O)Ph]₃ at a constant concentration of 0.01 mM with different concentrations of G6 (4-trifluoromethylbenzonitrile), ranging from 0 mM to 0.47 mM. (b). The absorbance changes of F[31A1-[P(O)Ph]₃ upon the addition of G6. The red solid line was obtained from the non-linear curve fitting. The association constant (K_a) between F[31A1-[P(O)Ph]₃ and G6 was estimated to be $(6.253 \pm 0.377) \times 10^3 \text{ M}^{-1}$. (c). UV-Vis absorption spectra of complex F[31A1-[P(O)Ph]₃ @ G6 with different molar ratios in water while [F[31A1-[P(O)Ph]₃] + [G6] = 0.10 mM. (d). Job plots of the complex F[31A1-[P(O)Ph]₃ @ G6 showing a 1:1 stoichiometry between F[31A1-[P(O)Ph]₃ and G6 by plotting the absorbance differences at 265 nm (a characteristic absorption peak of F[31A1-[P(O)Ph]₃) against the mole fraction of G6.

Figure R37. (a). UV-Vis absorption spectra of F[31A1-[P(O)Ph]₃ at a constant concentration of 0.01 mM with different concentrations of G7 (4-ethylbenzonitrile), ranging from 0 mM to 0.76 mM. (b). The absorbance changes of F[31A1-[P(O)Ph]₃ upon the addition of G7. The red solid line was obtained from the non-linear curve fitting. The association constant (K_a) between F[31A1-[P(O)Ph]₃ and G7 was estimated to be $(6.446 \pm 0.339) \times 10^3 \text{ M}^{-1}$. (c). UV-Vis absorption spectra of complex F[31A1-[P(O)Ph]₃ @ G7 with different molar ratios in water while [F[31A1-[P(O)Ph]₃] + [G7] = 0.10 mM. (d). Job plots of the complex F[31A1-[P(O)Ph]₃ @ G7 showing a 1:1 stoichiometry between F[31A1-[P(O)Ph]₃ and G7 by plotting the absorbance differences at 265 nm (a characteristic absorption peak of F[31A1-[P(O)Ph]₃) against the mole fraction of G7.

Figure R38. (a). UV-Vis absorption spectra of F[3]A1-[P(O)Ph]₃ at a constant concentration of 0.01 mM with different concentrations of G8 (4-butylbenzotrile), ranging from 0 mM to 0.74 mM. (b). The absorbance changes of F[3]A1-[P(O)Ph]₃ upon the addition of G8. The red solid line was obtained from the non-linear curve fitting. The association constant (K_a) between F[3]A1-[P(O)Ph]₃ and G8 was estimated to be $(8.065 \pm 0.491) \times 10^3 \text{ M}^{-1}$. (c). UV-Vis absorption spectra of complex F[3]A1-[P(O)Ph]₃ @ G8 with different molar ratios in water while $[\text{F[3]A1-[P(O)Ph]}_3] + [\text{G8}] = 0.10 \text{ mM}$. (d). Job plots of the complex F[3]A1-[P(O)Ph]₃ @ G8 showing a 1:1 stoichiometry between F[3]A1-[P(O)Ph]₃ and G8 by plotting the absorbance differences at 265 nm (a characteristic absorption peak of F[3]A1-[P(O)Ph]₃) against the mole fraction of G8.

Figure R39. (a). UV-Vis absorption spectra of F[31A1-[P(O)Ph]₃ at a constant concentration of 0.01 mM with different concentrations of G9 (3,4-dihydroxybenzointrile), ranging from 0 mM to 0.68 mM. (b). The absorbance changes of F[31A1-[P(O)Ph]₃ upon the addition of G9. The red solid line was obtained from the non-linear curve fitting. The association constant (K_a) between F[31A1-[P(O)Ph]₃ and G9 was estimated to be $(1.8087 \pm 0.0413) \times 10^4 \text{ M}^{-1}$. (c). UV-Vis absorption spectra of complex F[31A1-[P(O)Ph]₃ @ G9 with different molar ratios in water while [F[31A1-[P(O)Ph]₃] + [G9] = 0.10 mM. (d). Job plots of the complex F[31A1-[P(O)Ph]₃ @ G9 showing a 1:1 stoichiometry between F[31A1-[P(O)Ph]₃ and G9 by plotting the absorbance differences at 265 nm (a characteristic absorption peak of F[31A1-[P(O)Ph]₃) against the mole fraction of G9.

Figure R40. (a). UV-Vis absorption spectra of F[31A1]-[P(O)Ph]₃ at a constant concentration of 0.01 mM with different concentrations of G10 (3-fluoro-4-hydroxybenzoxonitrile), ranging from 0 mM to 0.59 mM. (b). The absorbance changes of F[31A1]-[P(O)Ph]₃ upon the addition of G10. The red solid line was obtained from the non-linear curve fitting. The association constant (K_a) between F[31A1]-[P(O)Ph]₃ and G10 was estimated to be $(1.4557 \pm 0.0515) \times 10^4 \text{ M}^{-1}$. (c). UV-Vis absorption spectra of complex F[31A1]-[P(O)Ph]₃ @ G10 with different molar ratios in water while [F[31A1]-[P(O)Ph]₃] + [G10] = 0.10 mM. (d). Job plots of the complex F[31A1]-[P(O)Ph]₃ @ G10 showing a 1:1 stoichiometry between F[31A1]-[P(O)Ph]₃ and G10 by plotting the absorbance differences at 265 nm (a characteristic absorption peak of F[31A1]-[P(O)Ph]₃) against the mole fraction of G10.

Figure R41. (a). UV-Vis absorption spectra of F[31A1-[P(O)Ph]₃ at a constant concentration of 0.01 mM with different concentrations of G11 (4-cyanobenzenesulfonamide), ranging from 0 mM to 0.64 mM. (b). The absorbance changes of F[31A1-[P(O)Ph]₃ upon the addition of G11. The red solid line was obtained from the non-linear curve fitting. The association constant (K_a) between F[31A1-[P(O)Ph]₃ and G11 was estimated to be $(1.3445 \pm 0.0373) \times 10^4 \text{ M}^{-1}$. (c). UV-Vis absorption spectra of complex F[31A1-[P(O)Ph]₃ @ G11 with different molar ratios in water while [F[31A1-[P(O)Ph]₃] + [G11] = 0.10 mM. (d). Job plots of the complex F[31A1-[P(O)Ph]₃ @ G11 showing a 1:1 stoichiometry between F[31A1-[P(O)Ph]₃ and G11 by plotting the absorbance differences at 265 nm (a characteristic absorption peak of F[31A1-[P(O)Ph]₃) against the mole fraction of G11.

Figure R42. (a). UV-Vis absorption spectra of F[31A1-[P(O)Ph]₃ at a constant concentration of 0.01 mM with different concentrations of G12 (4'-Hydroxy-4-biphenylcarbonitrile), ranging from 0 mM to 0.71 mM. (b). The absorbance changes of F[31A1-[P(O)Ph]₃ upon the addition of G12. The red solid line was obtained from the non-linear curve fitting. The association constant (K_a) between F[31A1-[P(O)Ph]₃ and G12 was estimated to be $(1.9324 \pm 0.0555) \times 10^4 \text{ M}^{-1}$. (c). UV-Vis absorption spectra of complex F[31A1-[P(O)Ph]₃ @ G12 with different molar ratios in water while [F[31A1-[P(O)Ph]₃] + [G12] = 0.10 mM. (d). Job plots of the complex F[31A1-[P(O)Ph]₃ @ G12 showing a 1:1 stoichiometry between F[31A1-[P(O)Ph]₃ and G12 by plotting the absorbance differences at 265 nm (a characteristic absorption peak of F[31A1-[P(O)Ph]₃) against the mole fraction of G12.

Figure R43. (a). UV-Vis absorption spectra of $F[31A1-[P(O)Ph]_3]$ at a constant concentration of 0.01 mM with different concentrations of **G13** (1,4-bis(4-cyanostyryl)benzene), ranging from 0 mM to 0.574 mM. (b). The absorbance changes of $F[31A1-[P(O)Ph]_3]$ upon the addition of **G13**. The red solid line was obtained from the non-linear curve fitting. The association constant (K_a) between $F[31A1-[P(O)Ph]_3]$ and **G13** was estimated to be $(2.1586 \pm 0.0763) \times 10^4 \text{ M}^{-1}$. (c). UV-Vis absorption spectra of complex $F[31A1-[P(O)Ph]_3 @ G13]$ with different molar ratios in water while $[F[31A1-[P(O)Ph]_3] + [G13]] = 0.10 \text{ mM}$. (d). Job plots of the complex $F[31A1-[P(O)Ph]_3 @ G13]$ showing a 1:1 stoichiometry between $F[31A1-[P(O)Ph]_3]$ and **G13** by plotting the absorbance differences at 265 nm (a characteristic absorption peak of $F[31A1-[P(O)Ph]_3]$) against the mole fraction of **G13**.

Figure R44. (a). UV-Vis absorption spectra of **F[31A1-[P(O)Ph]₃** at a constant concentration of 0.01 mM with different concentrations of **G14** (crisaborole), ranging from 0 mM to 0.88 mM. (b). The absorbance changes of **F[31A1-[P(O)Ph]₃** upon the addition of **G14**. The red solid line was obtained from the non-linear curve fitting. The association constant (K_a) between **F[31A1-[P(O)Ph]₃** and **G14** was estimated to be $(1.1505 \pm 0.0638) \times 10^4 \text{ M}^{-1}$. (c). UV-Vis absorption spectra of complex **F[31A1-[P(O)Ph]₃ @ G14** with different molar ratios in water while $[\text{F[31A1-[P(O)Ph]}_3] + [\text{G14}] = 0.10 \text{ mM}$. (d). Job plots of the complex **F[31A1-[P(O)Ph]₃ @ G14** showing a 1:1 stoichiometry between **F[31A1-[P(O)Ph]₃** and **G14** by plotting the absorbance differences at 265 nm (a characteristic absorption peak of **F[31A1-[P(O)Ph]₃**) against the mole fraction of **G14**.

Figure R45. (a). UV-Vis absorption spectra of F[31A1-[P(O)Ph]₃ at a constant concentration of 0.01 mM with different concentrations of G15 (alectinib), ranging from 0 mM to 0.665 mM. (b). The absorbance changes of F[31A1-[P(O)Ph]₃ upon the addition of G15. The red solid line was obtained from the non-linear curve fitting. The association constant (K_a) between F[31A1-[P(O)Ph]₃ and G15 was estimated to be $(1.6692 \pm 0.0465) \times 10^4 \text{ M}^{-1}$. (c). UV-Vis absorption spectra of complex F[31A1-[P(O)Ph]₃ @ G15 with different molar ratios in water while $[\text{F[31A1-[P(O)Ph]}_3] + [\text{G15}] = 0.10 \text{ mM}$. (d). Job plots of the complex F[31A1-[P(O)Ph]₃ @ G15 showing a 1:1 stoichiometry between F[31A1-[P(O)Ph]₃ and G15 by plotting the absorbance differences at 265 nm (a characteristic absorption peak of F[31A1-[P(O)Ph]₃) against the mole fraction of G15.

Responses to comments from Reviewer 3:

This is a submission with considerable merit, but also one that requires much work prior to publication. In essence the authors detail what appears to be a new phosphine oxide bridged rigid macrocycle and show it is quite selective in stabilizing crystalline complexes of substituted benzonitriles. The best result is obtaining a crystal structure of an alectinib, an FDA drug that according to the authors had never been subject to an X-ray diffraction analysis. This is a big deal and provides a small molecule complement to Makoto Fujita's award winning cage-based crystal structure determination approach. This is an aspect of the "story" that deserves more air time. It should drive acceptance of this work on its own.

Response:

We greatly appreciate your positive comments and helpful suggestions to improve the quality of our manuscript. These comments are all valuable and helpful for improving our article.

Q1: *Conversely, the premise that the present crystallization approach can some how inform or replace computational biological-substrate analyses strikes this reviewer as hype. Actually, a stronger word comes to mind, but it isn't appropriate for a professional document. Calculations are benchmarked to experiment and programs like Schroedinger are constantly recalibrating. There is almost no calibration with theory in the present submission. How do experiment and theory differ in their well-defined system. What are the errors in theory or computational structural analysis that their new receptor has unearthed?*

Response:

In terms of the term "Molecular docking", our original idea is presented as follows: "Molecular docking" is a great technology for computer-aided drug design. The crucial phase in the molecular docking procedure involves the meticulous preparation of the structures of both the protein receptor and guest molecules, which are often obtained from sources such as X-ray crystallography or protein structure databases. (*Mol. Cancer*, **2016**,15, 64; *Nat. Biomed. Eng.*, **2022**, 6, 1180-1195; *J. Hematol. Oncol.*, **2020**,13,141;

Nat. Commun. **2022**, 13, 6105.). Therefore, the molecular docking is closely related to the crystal structures. Just as reviewer 1 mentioned, in this work, our original idea was to mimic the molecular docking (and vice versa) to describe the specific binding of the host macrocyclic **F[3]A1-[P(O)Ph]₃** to benzonitrile derivatives based on co-crystallization experiments.

As reviewers suggested, "molecular docking" may not be appropriately used for this work. In the revised manuscript. In the revised manuscript, we changed "molecular docking" into "precise recognition" to describe the specific binding of host-guest co-crystallization studies and the title of the revised manuscript has been changed to: "Precise recognition of benzonitrile derivatives with supramolecular macrocycle of phosphorylated cavitand by co-crystallization method." Some parts of the manuscript have also been revised accordingly shown as follows:

1. We have revised the abstract section highlighted in yellow in the revised manuscript (page 1) as follows:

"Molecular docking serves to establish specific lock-and-key recognition between target proteins and drug molecules in the field of structure-based drug design. The importance of molecular docking in drug discovery lies in the precise recognition between potential drug compounds and their target receptors, which is generally based on the computational method. However, it will become quite interesting if the rigid cavity structure of supramolecular macrocycles can precisely recognize a series of guests with specific fragments by mimicking molecular docking through co-crystallization experiments, instead of the conventional computational method."

2. We have revised the introduction section of the main text highlighted in yellow in the revised manuscript (page 3) as follows:

"However, the study of the precise recognition of a series of guests with specific fragments in the solid phase as the molecular docking does with supramolecular macrocycles is rare in supramolecular chemistry. Therefore, if the rigid cavity structure of supramolecular macrocycles can precisely recognize a series of guests with specific fragments by mimicking molecular docking through co-crystallization experiments, not only spatial and energetic complementarity with the target guest molecule can be achieved, but also the precision and reliability of this specific binding capability can be greatly improved. "

3. We modified "molecular docking" to "precise recognition" and "molecularly docked" to "precisely recognized", respectively, in the revised manuscript. (page 1, lines 4, 12,

and 28; page 2, left, lines 1, and 3; page 2, right, lines 1, and 3; page 3, left, lines 1, 7, 11, 30, 32, and 37; page 4, left, lines 21; page 4, right, lines 8, and 10; page 5, right, lines 40, 41 and 50; page 6, left, line 11; page 6, right, lines 21, and 29; page 7, left, lines 2, and 4; page 8, left, lines 2, 12, and 18; page 8, right, line 9).

Q2: *The constant use of phrases like "will hold significant promise in simulating protein-targeted discovery methodologies" and "opens up possibilities for designing and optimizing novel drug candidates" have to be backed up with real results! Grandiose words like "perfect" and "remarkable" should also be purged.*

Response:

Thank you for your expertise and recommendations. Following your suggestions, the relevant statements have been removed from the manuscript. The significance of this work has been modified: "The research can open up exciting possibilities for determining the structures of drugs containing benzonitrile fragments." (page 3, left, line 39; page 5, right, line 66; page 6, right, line 22; page 7, left, line 8; page 8, left, line 22) in the revised manuscript. In addition, the words "perfect" and "remarkable" have been deleted in the revised manuscript.

Q3: *The authors should state their findings and express their opinions without inflating them through borrowed importance. The term pi-pi stacking, which refers only to geometry, should also be replaced by donor-acceptor, which constitutes a real interaction. The authors should also be honest in noting that there are thousands of host-guest complexes whose structure has been solved. So, only by relating their contribution to an actual unsolved problem and showing its solution can publication in Nature Communications be justified. The structure determination of drugs (as noted above) could provide this opportunity.*

Response:

Thank you for your expert recommendations. According to your suggestions, we have reviewed the literatures. The term "*u-u stacking interactions*" is employed to emphasize the relatively weak interaction between the host and guest (*Nat. Chem.* **2022**, 14, 1158 - 1164; *Angew. Chem. Int. Ed.* **2017**, 37, 11252-11257; *J. Am. Chem. Soc.* **2021**, 143, 973 - 982; *Angew. Chem. Int. Ed.* **2023**, 62, e202313696), whereas the term "*u-u*

stacking geometry" you mentioned emphasizes the way in which molecules are stacked (*Aggregate* **2023**, 4, e347; *J. Am. Chem. Soc.* **2016**, 138, 16315-16321; *J. Am. Chem. Soc.* **2015**, 137, 11656 - 11665; *J. Am. Chem. Soc.* **2010**, 132, 1738-1739). In this work, " π - π stacking interactions" is used to emphasize the weak form of interaction between the macrocycle **F[3]A1-[P(O)Ph]₃** and the guest molecule with the benzene ring, rather than the stacking mode. Your reference to "donor-acceptor interactions" is also a good choice for this work. According to your suggestion, " π - π stacking interactions" has been modified to "donor and receptor interactions" in some discussion in the revised manuscript. (page 5, right, line 28; page 6, right, line 17; page 8, left, line 8) .

REVIEWERS' COMMENTS

Reviewer #1 (Remarks to the Author):

Thank you for reconsidering of crystal structure models and checking on some of the points I have raised. I recommend the acceptance of this work.

Reviewer #2 (Remarks to the Author):

The authors have revised the manuscript according to the comments and suggestions of the reviewers, and the concerns have been basically addressed. So, its publication is suggested.

Reviewer #1:

Thank you for reconsidering of crystal structure models and checking on some of the points I have raised. I recommend the acceptance of this work.

Response:

We sincerely thank the reviewer for the positive comments on this work. And we are very much grateful for the insightful comments and suggestions that help us to improve the quality of this work.

Reviewer #2:

The authors have revised the manuscript according to the comments and suggestions of the reviewers, and the concerns have been basically addressed. So, its publication is suggested.

Response:

We thank the reviewer for the careful evaluation and acceptance of our revised manuscript. This is a great encouragement. We sincerely appreciate that the two reviewers can recognize our work and agree to publish it in Nature Communications.